# INTERP3D: CORRESPONDENCE-AWARE INTERPOLATION FOR GENERATIVE TEXTURED 3D MORPHING

**Xiaolu Liu**[1,2] **Yicong Li**[2*] **Qiyuan He**[2] **Jiayin Zhu**[2] **Wei Ji**[3] **Angela Yao**[2] **Jianke Zhu**[1,4*]

[1]State Key Lab of CAD & CG, Zhejiang University  [2]National University of Singapore
[3]Nanjing University  [4]Shenzhen Loop Area Institute

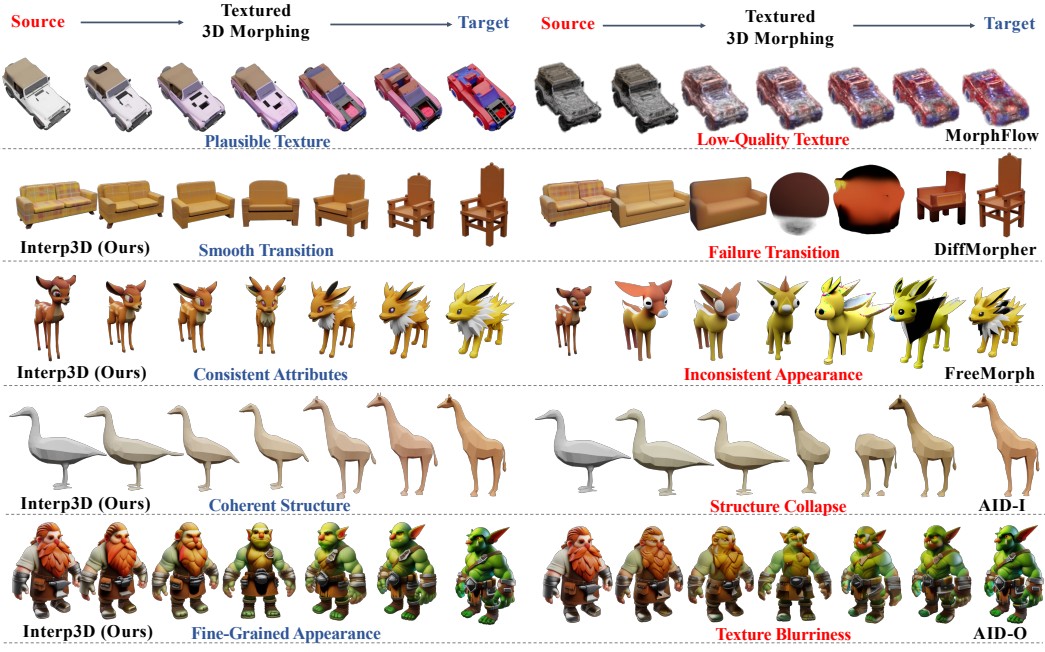

Figure 1: We propose **Interp3D**, a training-free approach for continuous and plausible textured 3D morphing, consistently surpassing prior approaches with better structure fidelity, plausible appearance, and transition smoothness. Zoom in to check the details.

## ABSTRACT

Textured 3D morphing seeks to generate smooth and plausible transitions between two 3D assets, preserving both structural coherence and fine-grained appearance. This ability is crucial not only for advancing 3D generation research but also for practical applications in animation, editing, and digital content creation. Existing approaches either operate directly on geometry, limiting them to shape-only morphing while neglecting textures, or extend 2D interpolation strategies into 3D, which often causes semantic ambiguity, structural misalignment, and texture blurring. These challenges underscore the necessity to jointly preserve geometric consistency, texture alignment, and robustness throughout the transition process. To address this, we propose **Interp3D**, a novel training-free framework for textured 3D morphing. It harnesses generative priors and adopts a progressive alignment principle to ensure both geometric fidelity and texture coherence. Starting from semantically aligned interpolation in condition space, Interp3D enforces structural consistency via SLAT (Structured Latent)-guided structure interpolation, and finally transfers appearance details through fine-grained texture fusion. For comprehensive evaluations, we construct a dedicated dataset, Interp3DData, with graded difficulty levels and assess generation results from fidelity, transition smoothness, and plausibility. Both quantitative metrics and human studies demonstrate the sig-

---

*Corresponding authors.

nificant advantages of our proposed approach over previous methods. Source code is available at https://github.com/xiaolul2/Interp3D.

# 1 INTRODUCTION

Textured 3D morphing aims to generate smooth and consistent transitions between two 3D assets (Lipman et al., 2022; Yang et al., 2025). The core is to preserve natural evolution while avoiding semantically meaningless artifacts or abrupt changes. By integrating both structure and appearance evolution, coherent transitions are essential to support generative tasks, including animation (Ren et al., 2024), editing (Alimohammadi et al., 2025), and motion tasks (Miao et al., 2025; Nag et al., 2025). In practice, textured 3D morphing is also crucial for effects visualization and filmmaking, such as character or biological evolution in games and morphological changes, enabling realistic and immersive visual narratives that meet the demand for continuous variations.

Nevertheless, achieving fidelity and plausible transitions remains challenging. Prior studies have partially addressed this issue. One major paradigm is traditional deformation-based approaches, which operate on point clouds or meshes (Yan et al., 2007; Eisenberger et al., 2021) to establish explicit geometric correspondences, and then compute a deformation trajectory to generate intermediate states (Aydınlılar & Sahillioğlu, 2021; Vyas et al., 2021; Zhan et al., 2024). However, relying on strict geometric alignment and consistent topology, these approaches are restricted to shape-only interpolation and neglect textures, often leading to ambiguous matching and unnatural deformations.

More recently, generative models have been widely developed (Sauer et al., 2023; Podell et al., 2023) as the basis prior for morphing in 2D. For image morphing, interpolations on noises (Shen et al., 2024), attention mechanisms (Zhang et al., 2024; Alimohammadi et al., 2025), and conditional features (He et al., 2024; Wang & Golland, 2023) have been investigated to achieve semantic or texture transitions, offering an inherent ability to generate and create visually detailed surface appearances. Extending these ideas to 3D generally follows two routes: (1) morphing in image space to guide the 3D process (Sun et al., 2024), or (2) adapting 2D strategies to 3D generative models (Yang et al., 2025). However, both are essentially 2D-native: the former suffers from view inconsistency and error accumulation, while the latter ignores structural correspondences and struggles with scale or topology variations. Consequently, they often produce distorted or implausible transitions with blurred geometry and unstable textures, just as shown in Figure 1.

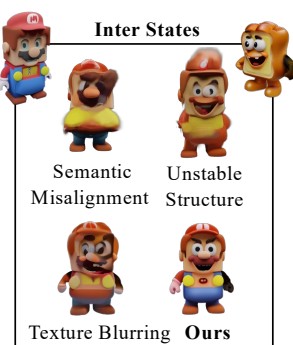

Considering the above limitations, our key insight is to couple generative priors with reliable 3D correspondences for faithful morphing. However, such a combination is non-trivial, particularly when structural and semantic gaps cause unstable correspondence. This motivated us to advocate a progressive three-stage alignment principle. Taking the morphing from Mario to Bread in Figure 2 as an example, *semantic alignment* acts as a high-level planner, establishing a conceptual map that ensures Mario's head matches with Bread's head, instead of its belly. *Structural alignment* then regularizes the deformation between matched parts, as semantic correspondence alone cannot handle large shape gaps, which require smooth and scale-consistent deformation to avoid collapse. Finally, based on the aligned structure, *texture alignment* transfers materials and re-synthesizes details to ensure the consistency between their outfit, avoiding blurry blends or texture popping.

Figure 2: Artifacts caused by missing correspondence alignments.

Motivated by the above discussions, we propose **Interp3D**, a training-free approach that instantiates the progressive alignment principle based on generative priors for textured 3D morphing. Interp3D refines the source-target correspondences across three perspectives. Firstly, at the 2D condition level, we establish the semantic-aligned condition interpolation to ensure that the global context of the source and target is meaningfully matched. Then, we leverage the strong generative prior of the structured latent features from the 3D foundation model to maintain the geometric correspondences for structure generation, where the fused attention interpolation with dynamic patch matching is applied to promote coherent structural evolution. Finally, we retrieve the source and target features at their corresponding locations to apply weighted fusion so that reasonable appearance and fine details can be transferred at the texture level. Such a progressive strategy allows Interp3D to effectively

address the semantic ambiguity, geometric inconsistency, and texture blurring during the generation process, producing morphing trajectories with both structural fidelity and textural coherence.

For comprehensive evaluations, we construct a dedicated dataset, Interp3DData, with graded difficulty levels to assess from the fidelity, transition smoothness, and plausibility perspectives. Experimental results show that our Interp3D achieves consistently superior performance in both structural and perceptual quality. Our main contributions can be summarized as follows:

(1) We analyze the challenges of existing textured 3D morphing paradigms and highlight the necessity to couple correspondence with the 3D generative priors for structural and textural consistency.

(2) We propose Interp3D, a training-free, correspondence-aware morphing framework that integrates progressive alignment into the 3D generation process, preserving faithful morphing through three stages: Semantic, Structural, and Texture alignment.

(3) We curate Interp3DData, a benchmark dataset for textured 3D morphing categorized into three difficulty levels, on which our method achieves superior performance over prior baselines, especially on the most challenging hard cases.

## 2 RELATED WORK

**3D Morphing.** Morphing in 3D aims to generate smooth and consistent transitions between different shapes and textures. Traditionally, geometric methods operate directly on 3D representations through explicit interpolations or deformation (Tam et al., 2012; Yan et al., 2007), which can be broadly divided into manifold-based and deformation field approaches. The former (Heeren et al., 2012; Kilian et al., 2007) formulates interpolation as geodesic paths on a recovered shape manifold, while the latter (Eisenberger et al., 2019; 2021) directly estimates transformations between shapes under isometric assumptions. which are limited by the need for strict vertex correspondence and often struggle with shapes with varying topology. Based on neural radiance fields, MorphFlow (Tsai et al., 2022) formulates volumetric interpolation via optimal transport to synthesize view-consistent intermediate states. Later, interpolation in the latent feature space of generation model provides a way to achieve morphing (Achlioptas et al., 2018; Groueix et al., 2018). More recent works extend this idea beyond geometry. L4GM (Ren et al., 2024) interpolates RGB frames for 4D dynamic reconstruction at the image level. Yang et al. (2025) further introduces the first regenerative 3D morphing method built upon generative 3D models. Later, Yin et al. (2025) achieves interpolation via optimal transport barycenter on condition features. Inspired by these advances, we enable flexible generation of intermediate results with richer structural details and more realistic textures.

**3D Generative Models.** Initially, Generative Adversarial Networks (GANs) (Wu et al., 2016; Gao et al., 2022) are used for 3D generation with simple structures. Later approaches rely on diffusion models (Ho et al., 2020) with different 3D representations, such as point clouds (Nichol et al., 2022), meshes (Liu et al., 2023), Radiance Fields (Hong et al., 2024), and 3D Gaussians (Lan et al., 2025). Previous Score Distillation Sampling (Poole et al., 2022; Lin et al., 2023; Zhu et al., 2025) distills 3D information from 2D diffusion models. In spite of the encouraging results, these methods often suffer from limited fidelity and inefficient optimization. For high quality and efficient 3D generation, recent methods encode 3D data into a compact latent space with Variational Autoencoders (VAEs) and train diffusion models on these latents for scalable generation (Chen et al., 2025a; Lan et al., 2024). Among them, Trellis (Xiang et al., 2025) constructs a versatile structured latent space that can be decoded into various 3D representations. We utilize Trellis as the 3D diffusion prior, which contains structure and semantic latent space for correspondence-aware interpolation.

**Interpolation in Generative Models.** Interpolation in generative models has been widely explored for the tasks of morphing (Lin et al., 2025; Cao et al., 2025; Shen et al., 2024) and editing (Alimohammadi et al., 2025) in image and video generations (He et al., 2025; Liao et al., 2025). Previously, latent space interpolation has been studied in early generative models such as GANs (Fish et al., 2020; Park et al., 2020) and VAEs (Kingma & Welling, 2014), but their limited generalization restricts them to cases with simple semantics or highly similar structures. Recent advances in diffusion models have greatly improved interpolation for image morphing. For instance, AID (He et al., 2024) introduces attention-based interpolation. IMPUS (Yang et al., 2024) and DiffMorpher (Zhang et al., 2024) refine text embeddings with LoRA fine-tuning (Hu et al., 2022) for smoother transitions. Later, FreeMorph (Cao et al., 2025) achieves tuning-free image morphing with controllable

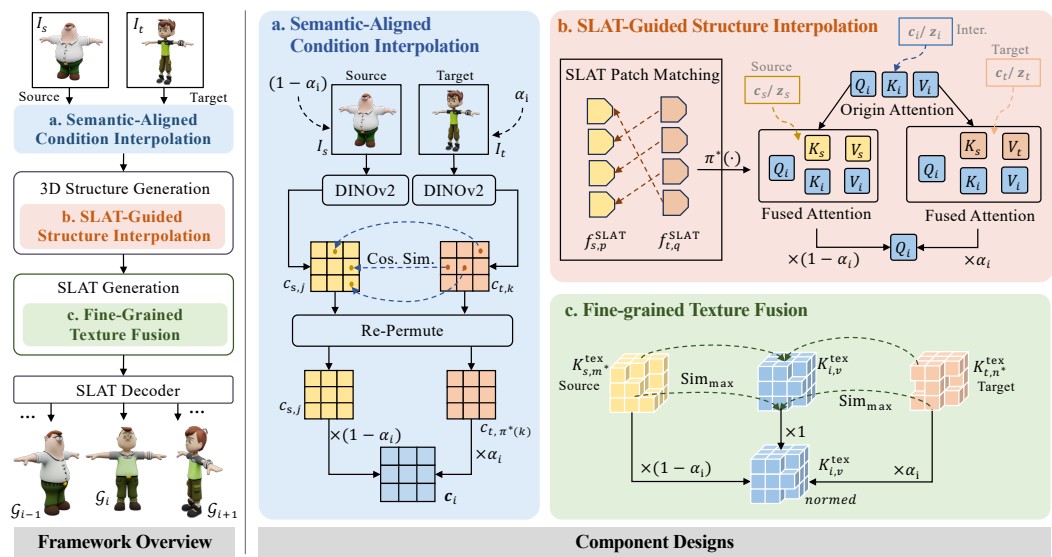

Figure 3: **Pipeline Overview.** The left presents the overall framework. The right highlights component designs. Based on the 3D generation prior, the interpolation is progressively enhanced from three aspects: **(a) Semantic-Aligned Condition Interpolation**, **(b) SLAT-Guided Structure Interpolation** in structure generation, and **(c) Fine-Grained Texture Fusion** for appearance refinement.

slerp interpolation. Despite extensive explorations on 2D space, interpolation in 3D space remains underexplored, in which the requirements for structure and semantic alignments are less explored in 2D morphing.

## 3 PRELIMINARIES

**TRELLIS Structure.** We adopt the pre-trained TRELLIS (Xiang et al., 2025) as our 3D generative prior, which encodes a unified Structural Latent (SLAT) representation for versatile decoding. Guided by embedded image conditions $\mathbf{c}$, TRELLIS operates in two diffusion stages: (1) In Stage-1 *Structure Generation*, it predicts the set of active voxel positions $\{p_v\}_{v=1}^{\mathcal{V}} \in \{0, 1, \dots, N-1\}^3$, where $N$ denotes the grid resolution and $\mathcal{V}$ is the number of active voxels, (2) then in Stage-2 *SLAT Generation*, building on the geometry, it recovers the texture-aware SLAT feature $\mathbf{z} = (z_v, p_v)_{v=1}^{\mathcal{V}}$, where each latent $z_v \in \mathbb{R}^C$ encodes fine-grained appearance at voxel $p_v$.

Finally, the SLAT decoder maps $\mathbf{z}$ to its corresponding 3D Gaussians $\mathcal{G}$. Both generative stages are built upon rectified flow transformers (Lipman et al., 2022), which iteratively denoise latent codes into clean structural and textural representations.

**Attention Interpolation.** In generative transformers, latent tokens are projected into query ($Q$), key ($K$), and value ($V$) spaces within each block to calculate attention. In tasks of generative morphing, attention-based interpolation is widely adopted (Zhang et al., 2024; Shen et al., 2024) to fuse the features from the source and target during the generation process. Given the $i$-th sample with interpolation ratio $\alpha_i \in [0, 1]$, the interpolated attention is defined as:

$$\text{InterpAttn}(Q_i, K_i, V_i) = \text{Attn}(Q_i, (1 - \alpha_i)K_s + \alpha_i K_t, (1 - \alpha_i)V_s + \alpha_i V_t), \quad (1)$$

where $K_i$ and $V_i$ are replaced by the interpolated source and target $\{K_s, K_t, V_s, V_t\}$. This formulation is a direct linear combination of source and target features within the attention mechanism.

## 4 METHODOLOGY

Given the source and target image prompts $I_s$ and $I_t$, our goal is to generate the sequence of textured 3D assets $\mathcal{S} = \{\mathcal{G}_i\}_{i=0}^{L-1}$ with smooth and plausible transitions, in which $L$ represents the length of the sequence. $G_0$ is generated from the source $I_s$, and $G_L$ is from the target $I_t$. As shown in Figure 3,

our Interp3D is designed to progressively enforce the aligned interpolation from three complementary perspectives. Firstly, we establish the correspondences between the source and target condition embeddings (Section 4.1) with semantic-aligned condition interpolation. Then, for plausible structure learning, the SLAT-Guided Structure Interpolation (Section 4.2) extends this alignment into 3D structure space. Finally, the Fing-Grained Texture Fusion (Section 4.3) bidirectionally aggregates source and target appearance details, ensuring coherent and realistic surface appearance.

## 4.1 SEMANTIC-ALIGNED CONDITION INTERPOLATION.

In morphing tasks, interpolation in the condition space is often used to guide the denoising process in diffusion models. While effective in 2D settings, this naive strategy easily ignores semantic correspondences in 3D generation, leading to feature mixing and visual artifacts. To enable faithful morphing, it is thus crucial to enforce semantic alignment between source and target conditions.

Given the input source and target images $I_s$ and $I_t$, the embedded conditions $\mathbf{c}_s$ and $\mathbf{c}_t$ are extracted by DINOv2 (Oquab et al., 2024), which provides strong representation ability and semantic information to serve as a reliable basis for correspondence estimation. Let $\mathbf{c}_s = \{c_{s,j}\}_{j=1}^{M}$ and $\mathbf{c}_t = \{c_{t,k}\}_{k=1}^{M}$ denote the patch-level embeddings with $M$ tokens each. We compute patch-level cosine similarities and formulate the correspondence estimation as an assignment problem:

$$\pi^{\star} = \arg \max_{\pi \in \mathcal{P}_M} \sum_{j,k=1}^{M} \frac{\langle c_{s,j}, c_{t,\pi(k)} \rangle}{\|c_{s,j}\| \, \|c_{t,\pi(k)}\|}, \tag{2}$$

where $\pi^{\star}$ denotes the optimal one-to-one mapping, and $\mathcal{P}_M$ is the set of all possible patch permutations over $M$ patches. The target embeddings $\mathbf{c}_t$ are then re-permuted according to $\pi^{\star}(\cdot)$, which yields a semantically aligned representation consistent with $\mathbf{c}_s$.

On top of this alignment, we perform the corresponding interpolation with the ratio coefficients $\{\alpha_i \in [0,1]\}_{i=0}^{L-1}$. The interpolation is performed as token-wise convex combinations of matched pairs, allowing each patch to evolve smoothly toward its semantic counterpart and providing more reliable guidance for the morphing:

$$\mathbf{c}_i = (1 - \alpha_i) \times \{c_{s,j}\}_{j=1}^{M} + \alpha_i \times \{c_{t,\pi^{\star}(k)}\}_{k=1}^{M}. \tag{3}$$

This alignment ensures that interpolation is performed between semantically consistent source–target token pairs. As a result, the intermediate conditions remain plausible and provide reliable guidance for the morphing trajectory, substantially reducing inconsistencies and preventing category-level mismatches from the outset.

## 4.2 SLAT-GUIDED STRUCTURE INTERPOLATION

While condition interpolation $\mathbf{c}_i$ provides semantically aligned guidance, it is limited to single-view 2D semantics and thus is hard to capture the spatial correspondence required for the intermediate 3D structural generation. To address this limitation, we introduce SLAT features from the source and target generation process as geometric guidance during the 3D generation process. As a versatile representation from structures and appearance, SLAT enables the construction of dynamic patch-level correspondences between the source and target, providing a principled mechanism for structure-aware interpolation.

**Dynamic Patch Correspondence.** As the denoising process follows a coarse-to-fine progression (Alimohammadi et al., 2025) early denoising steps predominantly recover global structural layouts, whereas later steps refine fine-grained details. Motivated by this, we design a dynamic patch correspondence mechanism that adapts its granularity across timesteps.

Specifically, we densify and project the sparse source and target SLAT features $f_s^{\text{SLAT}}$ and $f_t^{\text{SLAT}}$ into the same grid resolution as KV maps in stage-1 of structure learning. The grids are partitioned into patches with side length $s_t$ at denoise step $t$. Thus, the number of grid patches can be $G = \left\lceil \frac{N}{s_t} \right\rceil^3$. We then compute the cosine similarity between patch features rather than individual tokens. Only pairs with similarity above the threshold $\tau_0$ are retained for correspondence estimation, while the remaining unmatched patches are left to maintain the origin permutation. Formally, the optimal correspondence $\pi^{\star}$ can be obtained by:

$$\pi^{\star} = \arg\max_{\pi \in \mathcal{P}_G} \sum_{p,q=1}^{G} \text{sim}\left(f_{s,p}^{\text{SLAT}}, f_{t,\pi^{\star}(q)}^{\text{SLAT}}\right), \tag{4}$$

where $f_{s,p}^{\text{SLAT}}, f_{t,q}^{\text{SLAT}}$ are the corresponding SLAT patch features at source and target patch cell $g_s$, $g_t$. As denoising proceeds and the embeddings become increasingly reliable, the patch size $s_t$ is progressively reduced, ensuring robust coarse alignment in early stages and finer in later stages, adapting dynamically to the fidelity of the learned representations.

Given the estimated correspondence $\pi^{\star}(\cdot)$, we construct the permutation matrix $\mathcal{P}_{\pi}$ at the KV space in the transformer of the structure generation process, which is then applied to the target geometric $K_t, V_t$ maps to achieve the alignment with the source $K_s, V_s$:

$$\hat{K}_t^{\text{geo}}, \hat{V}_t^{\text{geo}} :\leftarrow P_{\pi^{\star}}(K_t^{\text{geo}}, V_t^{\text{geo}}). \tag{5}$$

This operation ensures that the source and target attention tokens are permuted accordingly to the guidance of the SLAT feature, providing the structure alignment for fused attention.

**Fused Attention Interpolation.** As illustrated in the Preliminary, the basic attention interpolation simply replaces the $K_i, V_i$ pairs with an interpolated version, ignoring the continuity with the generated intermediate states and the requirement for reliable source–target alignment.

Inspired by (He et al., 2024), we design the correspondence-aware fused attention interpolation based on the repermuted KV maps. For the self-attention in each transformer layer, we concatenate the $K_i, V_i$, which is constructed from the $i$-th interpolated condition $\mathbf{c}_i$ with the aligned $K_{\{s,t\}}, V_{\{s,t\}}$ from the source and target to perform the fused outer-interpolated attention with $Q_i$, which can be formulated as:

$$\begin{aligned} Q_i :\leftarrow & (1 - \alpha_i) * \text{SelfAttn}(Q_i, \left[K_s^{\text{geo}}, K_i^{\text{geo}}\right], \left[V_s^{\text{geo}}, V_i^{\text{geo}}\right] \\ & + \alpha_i * \text{SelfAttn}(Q_i, \left[\hat{K}_t^{\text{geo}}, K_i^{\text{geo}}\right], \left[\hat{V}_t^{\text{geo}}, V_i^{\text{geo}}\right]). \end{aligned} \tag{6}$$

This fused attention with SLAT-guided permutation enables each query to integrate the aligned structural information from both source and target conditions while preserving its own spatial cues, towards a more reasonable and structurally coherent 3D structure alignment.

## 4.3 Fine-Grained Texture Fusion

Based on the SLAT-guided structure learning, we further refine the texture representations to ensure smooth and consistent appearance transitions. As direct interpolation or binding textures to voxels often causes color distortion or blurring, especially when the source and target objects exhibit significant texture differences. Moreover, structural discrepancies between the source and target 3D objects result in different numbers of active voxels, making it impractical to directly adopt 2D interpolation strategies for texture learning based on different structures.

To address these issues, we introduce the fine-grained texture fusion strategy. The key idea is to perform bidirectional weighted aggregation of texture features from both source and target, which enables robust and consistent texture fusion regardless of voxel count differences.

Concretely, for each projected $j$-th token $K_{i,j}^{\text{tex}}$ from the $i$-th morphing process, we identify its most relevant counterparts from the source and target generation by selecting the features with the highest similarity, which can be formulated as follows:

$$m^* = \arg\max_m \text{sim}\left(K_{i,v}^{\text{tex}}, K_{s,m}^{\text{tex}}\right), \ n^* = \arg\max_n \text{sim}\left(K_{i,v}^{\text{tex}}, K_{t,n}^{\text{tex}}\right). \tag{7}$$

Here $m^*$ and $n^*$ represent the indices of the source and target tokens that are most similar to the $v$-th token in $K_i^{\text{tex}}$.

With the matched indices, each intermediate token is updated through a weighted aggregation of source, target, and its own feature. This approach ensures that the intermediate tokens not only inherit information from both endpoints but also retain their evolving identity during morphing, rather than collapsing into a simple linear blend. To maintain numerical stability and avoid feature magnitude drift, we further apply $\ell_2$ normalization:

$$K_{i,v}^{\text{tex}} :\leftarrow \frac{\|K_{i,v}^{\text{tex}}\|}{\|\tilde{K}_{i,v}^{\text{tex}}\|} \tilde{K}_{i,v}^{\text{tex}}, \ \text{where} \ \tilde{K}_{i,v}^{\text{tex}} = (1 - \alpha_i) K_{s,m^*}^{\text{tex}} + \alpha_i K_{t,n^*}^{\text{tex}} + K_{i,v}^{\text{tex}}. \tag{8}$$

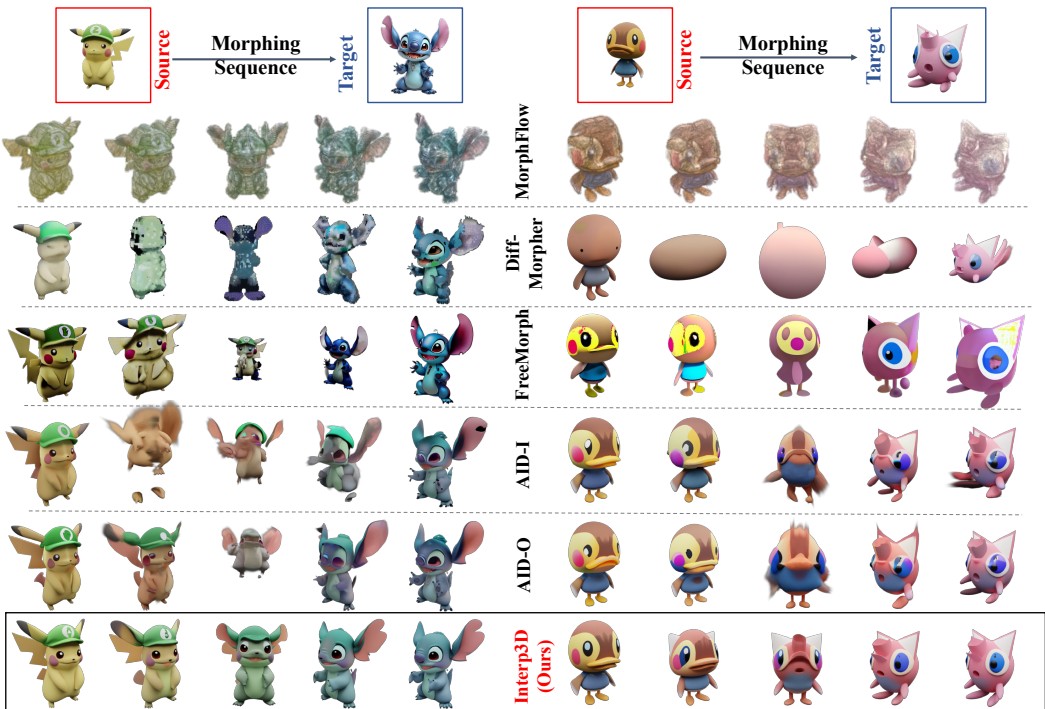

Figure 4: Qualitative Results. Our Interp3D achieves smooth and plausible 3D morphing.

The same update scheme is applied to the value tokens $V_{i,v}^{\text{tex}}$, ensuring consistency between keys and values throughout fusion. By jointly combining information from both source and target while retaining the intermediate feature, this design prevents collapsing toward either endpoint. Together, these choices allow each token to progressively shift toward source–target consensus while preserving its own structural cues, resulting in coherent texture alignment across varying voxel resolutions.

## 5 EXPERIMENTS

### 5.1 IMPLEMENTATION DETAILS

We utilize the publicly available Trellis (Xiang et al., 2025) as our 3D diffusion prior, which consists of 25 denoising steps for stage-1 structure generation and stage-2 latent construction, separately. The grid resolution is set to $N = 64$, and the SLAT dimension $C = 8$. Our approach can be integrated with this pipeline in a training-free manner. For dynamic patch correspondence, the maximum cube size is set to $4^3$, which is exponentially reduced as denoising progresses. Following AID (He et al., 2024), we adopt the Beta distribution with parameters $\beta = 5$ to sample non-uniform interpolation coefficients. It places more emphasis on intermediate interpolation states, leading to stable and balanced transition results. All experiments are conducted on a single NVIDIA RTX A5000 GPU.

### 5.2 EVALUATION BENCHMARKS

**Dataset Construction.** For comprehensive evaluations, we construct Interp3DData, a dataset consisting of 57 pairs categorized into three difficulty levels, including easy, medium, and hard, which are collected from the Objerverse-XL dataset (Deitke et al., 2023), TRELLIS repositories[1], and the Sketchfab website[2]. Each set contains 19 pairs. Categories include humans, objects, buildings, cartoon characters, and others. All selected cases comply with copyright and license requirements for research use. The difficulty levels are determined by geometric complexity, textural richness,

---

[1] https://github.com/microsoft/TRELLIS
[2] https://sketchfab.com

| Method | Easy | | | Mid | | | Hard | | | Average | | |
|--------|------|------|------|-----|------|------|------|------|------|---------|------|------|
| | FID↓ | PPL↓ | LPIPS↓ | FID↓ | PPL↓ | LPIPS↓ | FID↓ | PPL↓ | LPIPS↓ | FID↓ | PPL↓ | LPIPS↓ |
| MorphFlow | 101.36 | 2.79 | 0.111 | 107.57 | 2.96 | 0.156 | 105.71 | 2.92 | 0.187 | 104.88 | 2.89 | 0.151 |
| DiffMorpher | 160.93 | 4.18 | 0.088 | 177.45 | 4.16 | 0.113 | 170.26 | 4.92 | 0.183 | 169.54 | 4.42 | 0.128 |
| FreeMorph | 114.01 | 5.36 | 0.120 | 124.02 | 6.16 | 0.160 | 135.69 | 5.31 | 0.217 | 124.57 | 5.61 | 0.166 |
| AID-I | 84.35 | 2.92 | 0.072 | 84.64 | 3.10 | 0.104 | 94.64 | 3.57 | 0.159 | 87.88 | 3.20 | 0.112 |
| AID-O | 77.65 | 2.75 | 0.068 | 83.62 | 2.78 | 0.092 | **81.81** | 3.24 | 0.145 | 81.03 | 2.92 | 0.102 |
| **Interp3D (Ours)** | **70.79** | **2.42** | **0.059** | **83.58** | **2.37** | **0.079** | 82.54 | **2.62** | **0.119** | **78.97** | **2.47** | **0.086** |

Table 1: Quantitative evaluation on Interp3DData. We report FID, PPL, and LPIPS across three difficult levels and the average. Lower scores indicate better consistency and transition smoothness. The best is highlighted in **bold**.

and the structural discrepancy between the source and the target. We generate 7-frame morphing sequences to systematically assess the quality of transitions under varying levels of difficulty.

**Evaluation Metrics.** For comprehensive quality assessment, we evaluate the generated 3D sequences from three complementary aspects, including fidelity, transition smoothness, and plausibility. For fidelity, we employ the Fréchet Inception Distance (FID) (Heusel et al., 2017), which evaluates the distributional alignment between generated interpolations and the source–target data. To obtain a stable estimation, we render 1,000 views for the two selected intermediate frames, along with the corresponding source and target groups. For transition smoothness and consistency, we adopt Perceptual Path Length (PPL) (Karras et al., 2020) and the averaged Learned Perceptual Image Patch Similarity (LPIPS) (Zhang et al., 2018). PPL accumulates the normalized perceptual distances along the trajectory to capture transition regularity, and LPIPS quantifies local coherence between adjacent frames. Both metrics are averaged over 64 rendered views.

Beyond quantitative metrics, we include user studies in which participants provide ratings on fidelity, smoothness, plausibility, and overall quality, offering a more comprehensive evaluation.

## 5.3 QUANTITATIVE EVALUATIONS

**Baseline Selection.** As few methods are designed for textured 3D morphing, we select baselines from three representative directions, each with publicly available implementations:

(1) MorphFlow (Tsai et al., 2022) is an optimization method on neural radiance fields, which interpolates volumetric representations via optimal transport to enable view-consistent textured morphing.

(2) AID-I/O (He et al., 2024) is a basic investigation of inner and outer attention fused interpolation in 2D diffusion models. We extend its techniques into the 3D generation domain.

(3) DiffMorpher/ FreeMorph (Zhang et al., 2024; Cao et al., 2025) are image morphing methods that perform interpolation in 2D diffusion models. We integrate 2D morphing outputs into 3D generation pipelines, where the morphed images are served as inputs to synthesize sequential 3D shapes.

All these selected baselines establish a solid foundation for fair and comprehensive comparison with the most relevant existing methods.

**Evaluation of Fidelity and Transition Smoothness.** We evaluate fidelity and transition smoothness using FID, PPL, and LPIPS. As shown in Table 1, MorphFlow yields the 104.88 FID, reflecting poor volumetric generation quality. Despite its unusually low PPL and LPIPS, these values mainly result from diminished texture variation and oversimplified outputs, rather than genuinely smooth or coherent

| Method | Fidelity↑ | Smoothness↑ | Plausibility↑ | Overall↑ |
|--------|-----------|-------------|---------------|----------|
| DiffMorpher | 2.35% | 1.57% | 1.96% | 1.96% |
| FreeMorph | 9.02% | 12.16% | 10.20% | 10.46% |
| AID-O | 16.86% | 12.94% | 18.43% | 16.08% |
| MorphFlow | 17.65% | 23.14% | 11.37% | 17.39% |
| **Interp3D (Ours)** | **54.12%** | **50.20%** | **58.04%** | **54.12%** |

Table 2: User study on Interp3D against baselines. Higher values indicate stronger transition fidelity, smoother morphing, and better visual plausibility.

transitions. DiffMorpher and FreeMorph, which couple 2D morphing results, also perform poorly, producing PPL value 4.42 and 5.61 respectively due to inconsistencies when feeding 2D outputs to 3D models. When apply AID-I/O on 3D diffusion models, they achieve more stable geometry but still fail to preserve texture fidelity, leading to higher LPIPS. Conherently with the visual examples in Figure 4, our method achieves 78.97 FID score and 0.086 LPIPS, demonstrating the necessity of employing the combination of progressive alignment with 3D generative priors.

| Method | Easy | | | Mid | | | Hard | | | Average | | |
|---|---|---|---|---|---|---|---|---|---|---|---|---|
| | FID↓ | PPL↓ | LPIPS↓ | FID↓ | PPL↓ | LPIPS↓ | FID↓ | PPL↓ | LPIPS↓ | FID↓ | PPL↓ | LPIPS↓ |
| Initial Condition Interp. | 79.14 | 3.02 | 0.074 | 86.87 | 3.25 | 0.109 | 90.64 | 3.47 | 0.157 | 85.55 | 3.25 | 0.113 |
| + Semantic Align. | 75.09 | 2.75 | 0.067 | 86.82 | 2.98 | 0.101 | 88.63 | 3.24 | 0.146 | 83.51 | 2.99 | 0.105 |
| + Structure Interp. | 73.81 | 2.67 | 0.066 | 84.64 | 2.61 | 0.088 | 86.42 | 3.21 | 0.143 | 81.62 | 2.83 | 0.098 |
| + Texture Fusion | **70.79** | **2.42** | **0.059** | **83.58** | **2.37** | **0.079** | **82.54** | **2.62** | **0.119** | **78.97** | **2.47** | **0.086** |

Table 3: Ablation study on the progressive correspondence-aware component design across different difficulty levels. "Semantic alin." and "Structure interp." denote our semantic-aligned condition interpolation and SLAT-guided structure interpolation modules. Best results are highlighted in **bold**.

**User Study.** To evaluate perceptual plausibility and visual preference, we conducted a user study with 30 volunteers. Each participant was asked to assess 15 morphing cases by selecting the most convincing results among different methods. The proportion of selections for each method is reported in Table 2. The evaluation focused on three aspects: transition smoothness, structural consistency, and textural plausibility of the generated sequences. Interp3D is preferred by volunteers with overall probability 54.12% for its plausible transition sequences and preserving fine-grained appearance details in intermediate states.

## 5.4 VISUALIZED EVALUATIONS

Figure 1 and Figure 4 showcase representative morphing results. Interp3D produces smooth and semantically coherent transitions, with well-preserved geometry and fine-grained textures. In contrast, baselines often exhibit artifacts such as blurred textures, structural collapse, or inconsistent intermediate frames. These qualitative comparisons highlight the advantage of our correspondence-aware progressive design in generating visually plausible morphing trajectories. Additional visualizations are provided in the supplementary materials.

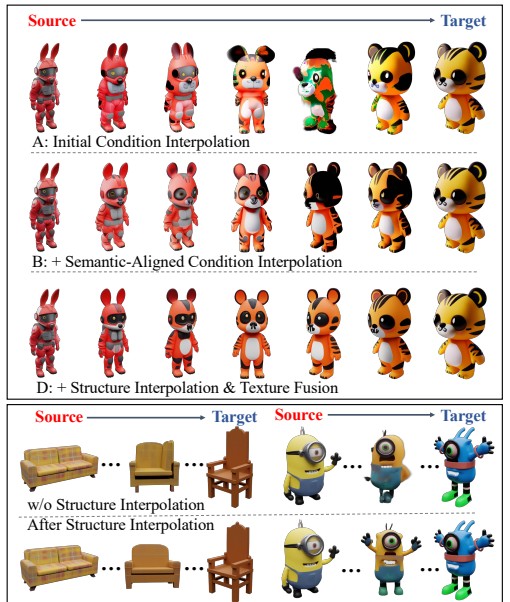

Figure 5: Visualized analysis for the effects of each component design.

## 5.5 ABLATION STUDY

In Table 3 and Figure 5, we analyze the effectiveness of our designs through both quantitative metrics and visualization. Based on Table 3, each progressively aligned component brings consistent improvements across all difficulty levels. Semantic alignment improve the consistency, espicially for ease cases with +4.06 FID reduced, highlighting its role in establishing meaningful correspondences. Structure interpolation further imporove the fidelity. Finally combining fine-grained texture fusion yields the best overall performance, especially for hard cases, with +0.59 PPL promotion and +0.024 in LPIPS.

In Figure 5 for the red rabbit–to–tiger case, the semantic-aligned condition interpolation effectively corrects semantic mismatches and sharpens structural boundaries. As shown in the below example, incorporating SLAT-guided structure interpolation strengthens structural consistency and alleviates posture-induced ambiguities, effectively handling cases like the structural gap between sofa and chair or the blurred hand poses in the minions transition.

## 6 CONCLUSION

In this work, we propose Interp3D, a correspondence-aware interpolation framework for generative textured 3D morphing. We design a progressive scheme to enforce alignment throughout the generative process, from semantically-aligned condition interpolation to SLAT-guided struc-

ture interpolation, and finally fine-grained texture fusion. This ensures both geometric fidelity and detail-preserving texture transitions. For evaluation, we constructed the dataset Interp3DData across difficulty levels, with performance assessed in consistency, transition smoothness, and plausibility. Results demonstrate that Interp3D outperforms existing baselines, delivering more coherent and elegant 3D transitions. We hope that our work can inspire future research in the field of 3D generation and beyond. A promising future direction lies in handling cases that share little semantic relevance, as building reliable correspondences for plausible transitions under such conditions remains a significant challenge.

## ETHICS STATEMENT

This work focuses on generative 3D morphing with textured outputs. All experiments are conducted on publicly available datasets, which contain 3D assets released under appropriate usage terms. Our method does not involve personally identifiable information, sensitive data.

For the user study, volunteers were recruited with informed consent. All studies were carried out in accordance with ethical guidelines to ensure participants were treated fairly and respectfully, with no personally identifiable information collected. The user feedback was used solely for research purposes.

Our method is intended for academic research in areas such as animation, creative design, and interactive editing. We acknowledge that generative techniques could be misused to produce misleading or harmful content, and therefore we emphasize responsible use of our models and results in alignment with community ethical standards.

## REPRODUCIBILITY STATEMENT

We are committed to ensuring the reproducibility of our work. All datasets used in this paper are publicly available, and detailed dataset sources are provided in the appendix. The model architectures, dataset settings, and experiment configurations are documented in the main text or supplementary materials. We provide the Interp3DData here, and our code is publicly available.

## ACKNOWLEDGEMENT

This work is supported by the National Natural Science Foundation of China under Grant No.62376244. It is also supported by the Information Technology Center and State Key Lab of CAD&CG, Zhejiang University. We would like to acknowledge that computational work involved in this research is partially supported by NUS IT's Research Computing group using grant number NUSREC-HPC-00001.

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

# A APPENDIX

In this Appendix, we provide more details on our implementation, experiments, visualization, and analysis as follows:

## A.1 INTERP3D GENERALIZATION ABILITY ANALYSIS.

Interp3D is designed as a model-agnostic concept. The key design is that progressive 3D correspondence modeling guided by native feature cues from the source and target generation process, which can be applied for different 3D generation models.

To validate this, we implemented Interp3D across two additional 3D generation baselines, LN3Diff (Lan et al., 2024) and 3DTopia-XL (Chen et al., 2025b). Based on the new baselines, we first perform semantic-aware condition interpolation, and then instantiate the structural correspondence using each model's native geometric embeddings (e.g., LN3Diff's 3D latent tokens, 3DTopia's primitive-level descriptors). These features provide stable geometric cues that let our structural alignment module plug in without modifying the backbone architectures. Finally, we apply our texture-refinement on their generative pipelines to complete the progressive alignment.

As shown in Figure 6, both approaches produce smooth and plausible morphings, confirming that Interp3D is a principled, architecture design rather than a model-specific enhancement.

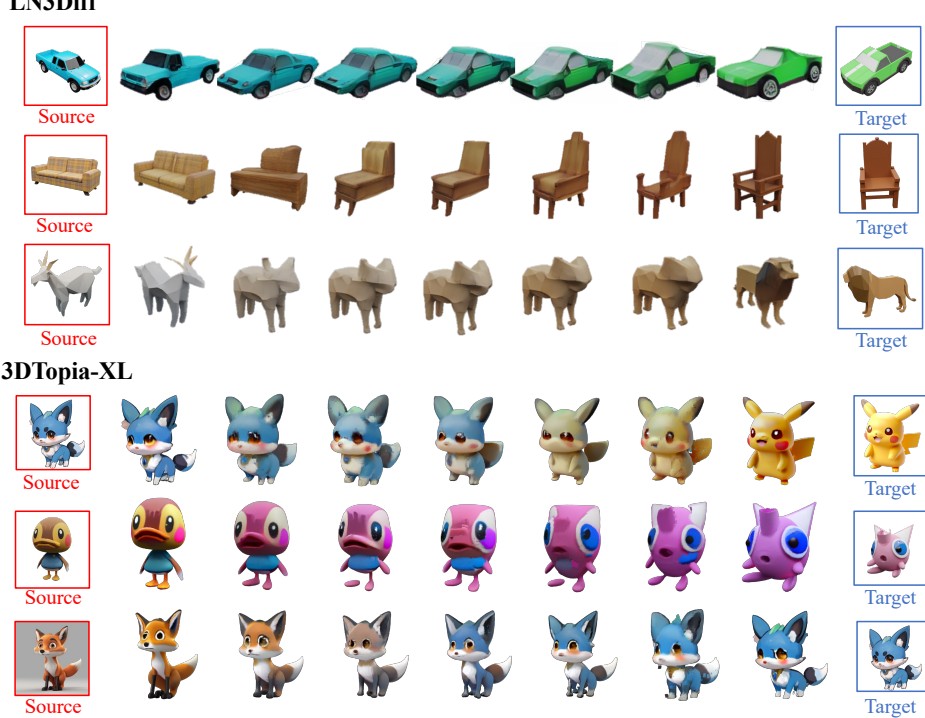

Figure 6: Visualization of Interp3D based on different 3D generation baselines.

## A.2 Implementation and Baseline Methods

### A.2.1 Pseudo Code

To better illustrate the overall workflow of Interp3D, we provide a pseudocode description of the entire pipeline. This step-by-step outline highlights the progressive alignment across semantic, structural, and textural levels, making the design and execution of our method clearer.

---

**Algorithm 1** Interp3D Framework with Progressive Diffusion Steps

---

**Require:** Source prompt $I_s$, target prompt $I_t$, interpolation ratios $\{\alpha_i\}_{i=0}^{L-1}$
**Ensure:** Morphing sequence $\mathcal{S} = \{\mathcal{G}_i\}_{i=0}^{L-1}$
 1: Extract embeddings $\mathbf{c}_s, \mathbf{c}_t$ with DINOv2
 2: Estimate semantic correspondences $\pi^\star$ by Eq. 2
 3: **Semantic-Aligned Condition Interpolation**
      Interpolate aligned embeddings for $\alpha_i$ by Eq. 3
 4: **for** each interpolation ratio $\alpha_i$ **do**
 5:    **for** diffusion step $t = 1$ to $T$ **do**
 6:        **SLAT-Guided Structure Interpolation**
          Project SLAT features into KV grid resolution
          Partition into patches, estimate dynamic correspondences by Eq. 4
          Permute target KV maps by Eq. 5
          Fuse attention with interpolated queries by Eq. 6
 7:        **Fine-Grained Texture Fusion**
          Find the most similar source/target tokens by Eq. 7
          Aggregate and normalize by Eq. 8

 8:    **end for**
 9:    Decode updated features into 3D Gaussians $\mathcal{G}_i$
10: **end for**
11: **Output:** Morphing sequence $\mathcal{S} = \{\mathcal{G}_i\}_{i=0}^{L-1}$

---

### A.2.2 Evaluation Details

We adopt three widely used metrics to evaluate the quality and consistency of textured 3D morphing.

**FID (Fréchet Inception Distance).** The Fréchet Inception Distance (FID) (Heusel et al., 2017) is a widely adopted metric for evaluating generative models. It measures the distance between the feature distributions of generated samples and real data, extracted by a pretrained Inception network. A lower FID score indicates that the generated outputs are closer to real samples in distribution.

In our textured 3D morphing task, the reference distribution is constructed from the source and target renderings, while the generated distribution is formed by 2 randomly selected intermediate states. This design evaluates whether the interpolated sequence lies within the perceptual manifold spanned by the endpoints, penalizing trajectories that drift away from both the source and target. For each state, we render 1000 images from multiple viewpoints to compute stable feature statistics. FID is then averaged across all intermediate steps, providing an overall measure of fidelity and consistency of the morphing trajectory.

**LPIPS (Learned Perceptual Image Patch Similarity).** LPIPS (Zhang et al., 2018) measures perceptual similarity between generated and reference renderings, using the pretrained VGG backbone as the feature extractor. For each case, we render 64 intermediate frames and compute the averaged LPIPS across all adjacent image pairs in the sequence to evaluate the transition smoothness, then average over all frames.

**PPL (Perceptual Path Length).** PPL evaluates the smoothness of interpolation trajectories by quantifying the perceptual difference between consecutive samples. Based on StyleGAN's definition (Sauer et al., 2023), which computes expected local smoothness under infinitesimal perturbations, here we adapt PPL to 3D morphing by normalizing the cumulative perceptual distance along the interpolation path with respect to the source–target perceptual distance. For each morphing trajectory, 64 rendered frames are sampled, the frame-to-frame perceptual distances are summed along

the path, and the overall average PPL across 64 views is reported. Smaller values indicate smoother and more consistent transitions.

### A.2.3 BASELINE METHODS

**MorphFlow (Tsai et al., 2022).** MorphFlow addresses the task of multiview image morphing, where the goal is to generate intermediate renderings between two sets of multiview images while ensuring cross-view consistency. It is based on a volumetric scene representation that jointly models geometry and appearance for each input set. The morphing process is formulated as an optimization that combines rigid transformation with optimal-transport interpolation under the Wasserstein metric. In our experiments, we firstly use BlenderNeRF [3] to obtain multi-view images together with their camera annotations, and generate the morphing results using the authors' open-source implementation [4] with default parameter settings.

**DiffMorpher (Zhang et al., 2024).** DiffMorpher tackles the problem of smooth image interpolation with diffusion models, whose latent spaces are otherwise unstructured and unsuitable for morphing. It captures the semantics of source and target images by fitting separate LoRAs and interpolates both the LoRA parameters and latent noises to generate coherent intermediate results. In our experiments, we use the open-sourced implementations [5] to generate the sequential images first, then the 3D generation model follows to lift the images into 3D sequences.

**FreeMorph (Cao et al., 2025).** FreeMorph is a tuning-free approach for instant image morphing, aiming to generate realistic and directional transitions between two images. It enhances diffusion models with guidance-aware spherical interpolation, which blends the key and value features in self-attention, and introduces a step-oriented variation trend to ensure controlled and consistent transitions. In our experiments, we use the open-sourced implementations [6] to generate the sequential images first, then the 3D generation model follows to lift the images into 3D sequences.

**AID (He et al., 2024).** AID (Attention Interpolation of Diffusion) improves conditional interpolation in diffusion models by directly modifying attention operations. The fused inner-interpolated attention (AID-I) fuses and interpolates keys and values before attention:

$$\text{Attn}_{\text{in}}(Q_i; \alpha_i) = \text{Attn}\Big(Q_i, [(1 - \alpha_i)K_s + \alpha_i K_t, K_i], [(1 - \alpha_i)V_s + \alpha_i V_t, V_i]\Big).$$

The fused outer-interpolated attention (AID-O) fuses and interpolates the outputs of two attentions:

$$\text{Attn}_{\text{out}}(Q_i; \alpha_i) = (1 - \alpha_i)\,\text{Attn}\Big(Q_i, [K_s, K_i], [V_s, V_i]\Big)$$
$$+ \alpha_i\,\text{Attn}\Big(Q_i, [K_t, K_i], [V_t, V_i]\Big).$$

In our experiments, we apply the open-source implementations [7] of both the AID-I and AID-O with basic condition interpolation in the structure generation process.

### A.3 MORE DETAILED EXPERIMENTS.

#### A.3.1 EVALUATION ON SEMANTIC AND STRUCTURE FIDELITY

We introduce two additional metrics that evaluate the 3D structural fidelity and semantic coherence:

**P-KID (Point-KID).** We extract PointNet (Qi et al., 2017) features from 3D point samples of the generated intermediate shapes and the source/target shapes, and then compute a Kernel Inception Distance between these distributions. Lower P-KID indicates that the intermediate shapes stay closer to the structural manifold spanned by the endpoints, thus reflecting better 3D geometric fidelity.

---

[3] https://github.com/maximeraafat/BlenderNeRF
[4] https://github.com/jimtsai23/MorphFlow
[5] https://github.com/Kevin-thu/DiffMorpher
[6] https://github.com/yukangcao/FreeMorph
[7] https://github.com/QY-H00/attention-interpolation-diffusion

**CLIP-Dis/ CLIP-Sim**. CLIP-based distance and similarity scores that quantify semantic alignment among the transition process. We use CLIP image features (Radford et al., 2021) to measure the averaged adjacent-frame distance (CLIP-Dis, lower is better) and cosine similarity (CLIP-Sim, higher is better), capturing whether semantic evolution is smooth and coherent.

As shown in the Table 4 below, we evaluate both our method and prior morphing approaches under these new metrics on the whole Interp3DData, and Interp3D consistently achieves better semantic continuity and geometric stability, confirming that our improvements extend beyond visual quality.

| Method | P-KID ↓ | CLIP-Dis ↓ | CLIP-Sim ↑ |
|---|---|---|---|
| DiffMorpher | 0.6796 | 0.1253 | 0.8747 |
| FreeMorph | 0.5352 | 0.1034 | 0.8966 |
| AID-I | 0.4857 | 0.0651 | 0.9349 |
| AID-O | 0.5060 | 0.0603 | 0.9397 |
| **Interp3D (Ours)** | **0.3961** | **0.0529** | **0.9471** |

Table 4: Quantitative evaluation on semantic and structure fidelity metrics.

| A | B | C | D | Easy FID↓ | PPL↓ | LPIPS↓ | Mid FID↓ | PPL↓ | LPIPS↓ | Hard FID↓ | PPL↓ | LPIPS↓ | Average FID↓ | PPL↓ | LPIPS↓ |
|---|---|---|---|---|---|---|---|---|---|---|---|---|---|---|---|
| ✓ | | | | 79.14 | 3.02 | 0.074 | 86.87 | 3.25 | 0.109 | 90.64 | 3.47 | 0.157 | 85.55 | 3.25 | 0.113 |
| ✓ | ✓ | | | 75.09 | 2.75 | 0.067 | 86.82 | 2.98 | 0.101 | 88.63 | 3.24 | 0.146 | 83.51 | 2.99 | 0.105 |
| ✓ | | ✓ | | 77.55 | 2.79 | 0.068 | 86.13 | 2.81 | 0.093 | 81.28 | 3.26 | 0.140 | 81.65 | 2.95 | 0.102 |
| ✓ | | | ✓ | 77.37 | 2.68 | 0.065 | 84.89 | 3.03 | 0.101 | 86.71 | 3.16 | 0.144 | 82.99 | 2.95 | 0.104 |
| ✓ | ✓ | ✓ | | 73.81 | 2.67 | 0.066 | 84.64 | 2.61 | 0.088 | 86.42 | 3.21 | 0.143 | 81.62 | 2.83 | 0.098 |
| ✓ | ✓ | | ✓ | 75.58 | 2.49 | 0.061 | 86.89 | 2.81 | 0.095 | 85.55 | 2.94 | 0.134 | 82.67 | 2.75 | 0.097 |
| ✓ | ✓ | ✓ | ✓ | **70.79** | **2.42** | **0.059** | **83.58** | **2.37** | **0.079** | **82.54** | **2.62** | **0.119** | **78.97** | **2.47** | **0.086** |

Table 5: Detailed Ablation study on component designs represented as A-D. Left side shows which components are enabled (✓). Best results are in **bold**.

### A.3.2 DETAILED ABLATIONS ON COMPONENT DESIGNS

We conduct a detailed ablation to verify the effects of each component design numerically. Table 5 presents the ablation results with four components: (A) initial condition interpolation, (B) semantic-aligned condition interpolation, (C) SLAT-guided structure interpolation, and (D) fine-grained texture fusion. It can be seen that the semantic-aligned condition interpolation contributes a lot to reducing the FID score, which means that it helps to improve the consistency of intermediate states with the source and target. Besides, the employment of fine-grained texture fusion produces lower LPIPS, reflecting its ability to refine local appearance details and enhance perceptual similarity.

### A.3.3 GAIN FROM THE BETA DISTRIBUTION.

As analyzed in AID (He et al., 2024) for 2D, linear interpolation with uniformly distributed interpolation coefficient $\alpha_i$ doesn't yield a uniformly spaced distribution with smooth transitions, which is also revealed during our experiments in 3D generation. Thus, we apply the concave Beta(5, 5) distribution with more shrinkage to the midpoint of the start and the end. As shown in the figure below, Beta sampling yields more smoothly transformed intermediate shapes than uniform sampling.

### A.4 INTERP3DDATA

**Source and Collection.** Our dataset, Interp3DData, is built from Objverse-XL dataset (Deitke et al., 2023), TRELLIS repositories[8], and the Sketchfab platform[9]. We carefully filtered assets based

---

[8] https://github.com/microsoft/TRELLIS
[9] https://sketchfab.com

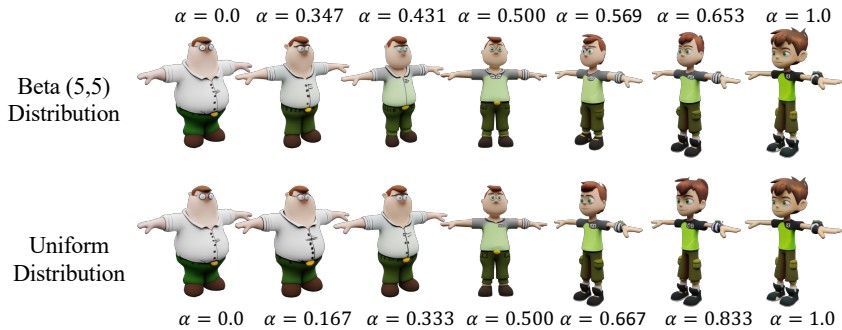

Figure 7: Visualized comparison between beta and uniform distribution for $\alpha_i$.

on quality, completeness, and license compliance for research usage. Each pair is selected to represent a meaningful semantic morphing scenario (e.g., two characters, two vehicles, two architectural forms), avoiding trivial rescalings or duplicates.

**Difficulty Levels.** To better capture the challenges in 3D morphing, we divide the dataset into three difficulty levels:

- **Easy:** Source and target have similar geometry and topology, or simple texture and appearance.

- **Medium:** Moderate geometric discrepancy or richer texture differences.

- **Hard:** Large variations or rich texture appearances, where both geometry and fine-grained textures differ drastically.

Each level contains 19 source–target pairs, for a total of 57 pairs.

As shown in Figure 8 below, we present a range of visualizations from our collected Interp3DData to enhance understanding of the dataset and the distinctions among its different levels.

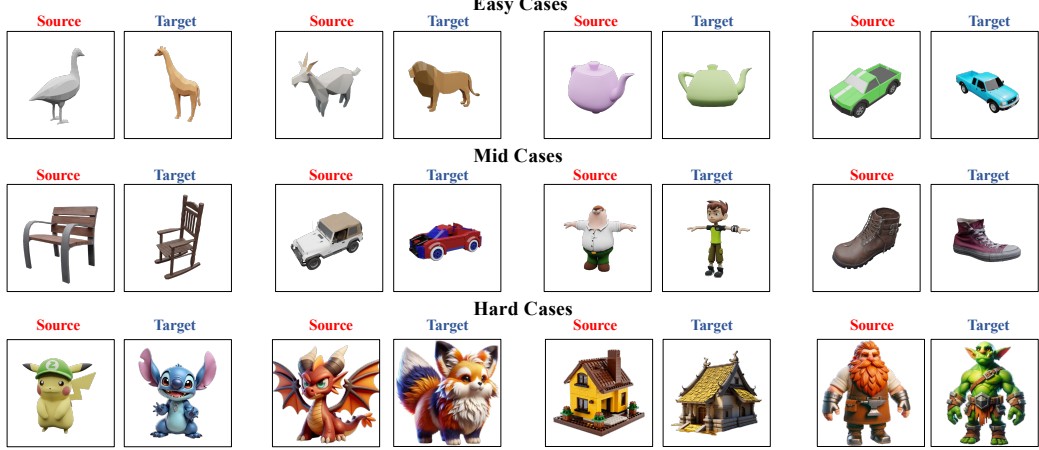

Figure 8: Examples of 3 difficulty levels in Interp3DData.

## A.5 MORE QUALITATIVE RESULTS.

Figure 10 presents more qualitative comparisons among Interp3D with previous approaches, where our method produces smoother and more coherent transitions, better preserving both structural fidelity and appearance details. As presented in Figure 11 and Figure 12, our method produces smooth and coherent transitions across a wide range of source–target pairs. These results further validate the effectiveness of our progressive alignment design in maintaining both structural fidelity and appearance consistency. Notably, the generated intermediates preserve fine details while avoiding abrupt artifacts, demonstrating the robustness of our approach under diverse scenarios.

## A.6 Applications and Future Analysis

**Content Creation and Design.** Interp3D can be directly applied to digital content production, where artists and designers often require smooth transformations between objects. For example, in animation and gaming, our framework can generate intermediate 3D assets between two key designs, reducing the manual effort of modeling transitional frames. In film production and advertising, morphing between different product shapes or styles allows creative visual effects without building separate 3D assets from scratch. Similarly, in AR/VR applications, smooth transitions between avatars or virtual objects can enhance immersion and user experience.

**Industrial and Interactive Applications.** Beyond creative fields, Interp3D supports practical industrial needs. In product prototyping, it allows designers to explore continuous shape and texture variations between initial concepts and final products, providing stakeholders with a clear visualization of design evolution. In e-commerce and virtual try-on systems, morphing between different product configurations (e.g., furniture colors, car models, or clothing textures) enables interactive previews for customers. In human–computer interaction, semantically aligned morphing can serve as an intuitive interface for adjusting 3D content along interpretable trajectories, such as gradually modifying size, style, or appearance.

## A.7 Failure Cases Analysis.

We do observe failures in extreme settings. As shown in Failure Case 1 of Figure 9, the source and target are semantically and structurally too far apart. In such cases, no reliable correspondence can be established, and the morph tends to collapse toward one endpoint, resulting in abrupt transitions and sudden changes rather than a smooth trajectory.

In Failure Case 2, we show an out-of-distribution example (e.g., sketched, flat 2D-like objects), where TRELLIS itself fails to reconstruct meaningful 3D geometry. Here, the morphing process degrades into collapsed geometry or heavy blur, since the 3D prior cannot provide a stable latent manifold to interpolate on.

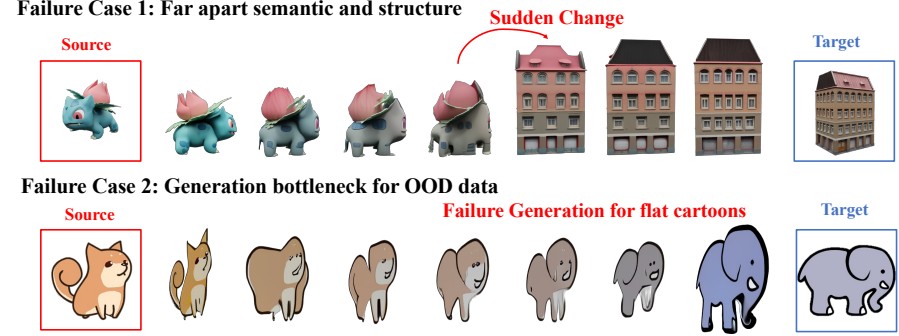

Figure 9: Failure case visualizations.

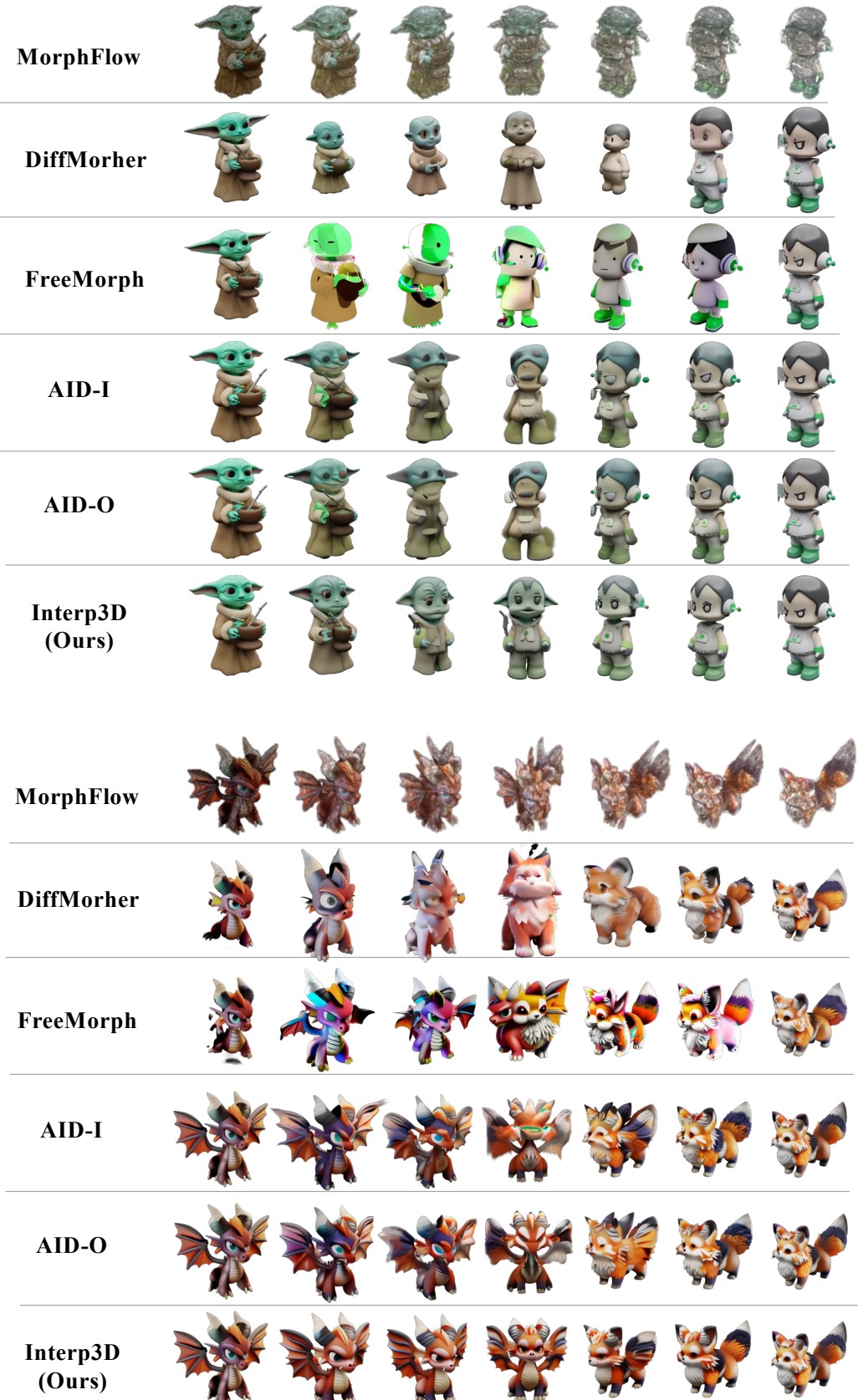

Figure 10: Qualitative comparisons of Interp3D with previous baselines.

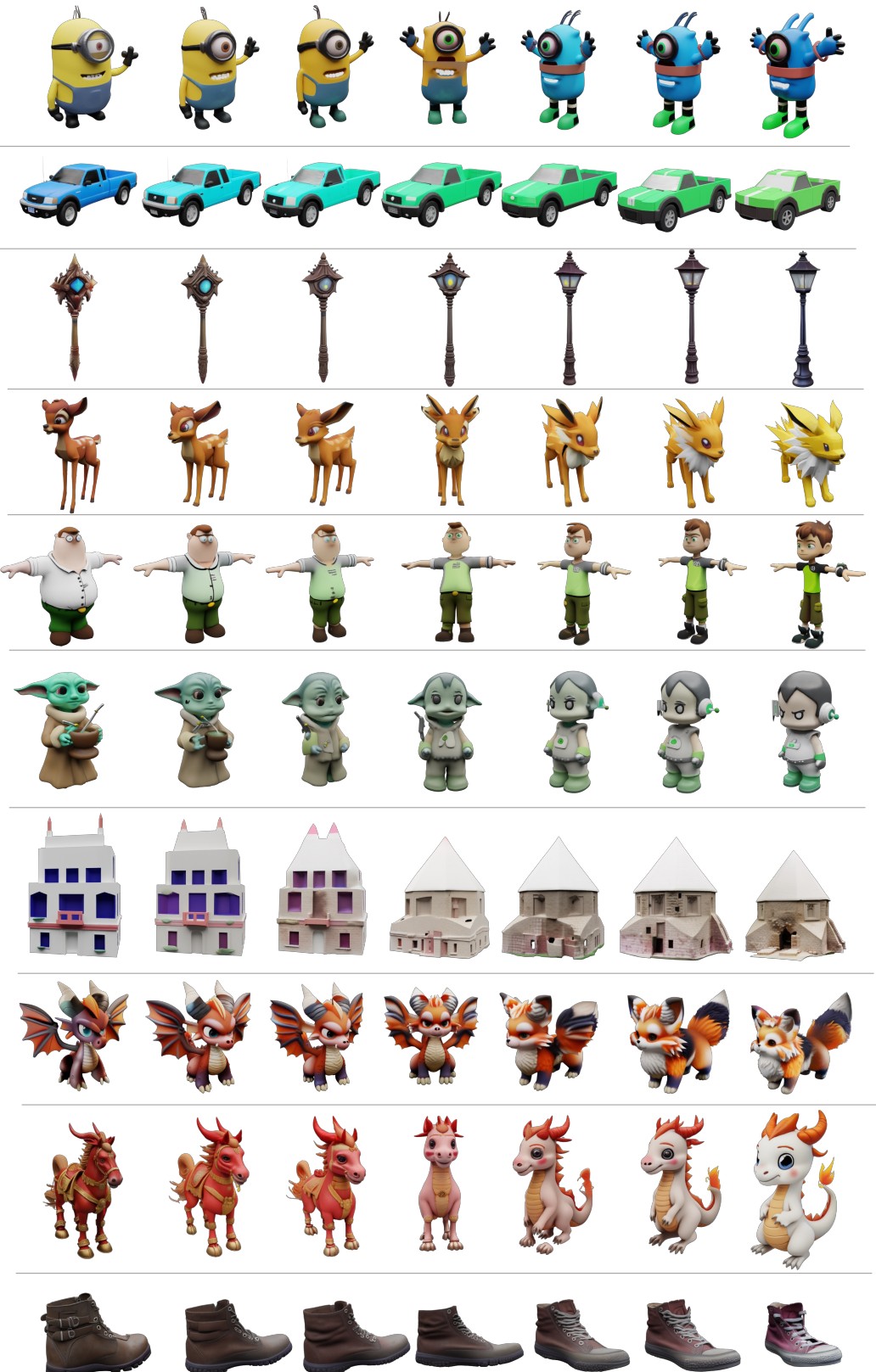

Figure 11: More qualitative results of Interp3D, which achieves fidelity and smooth transitions.

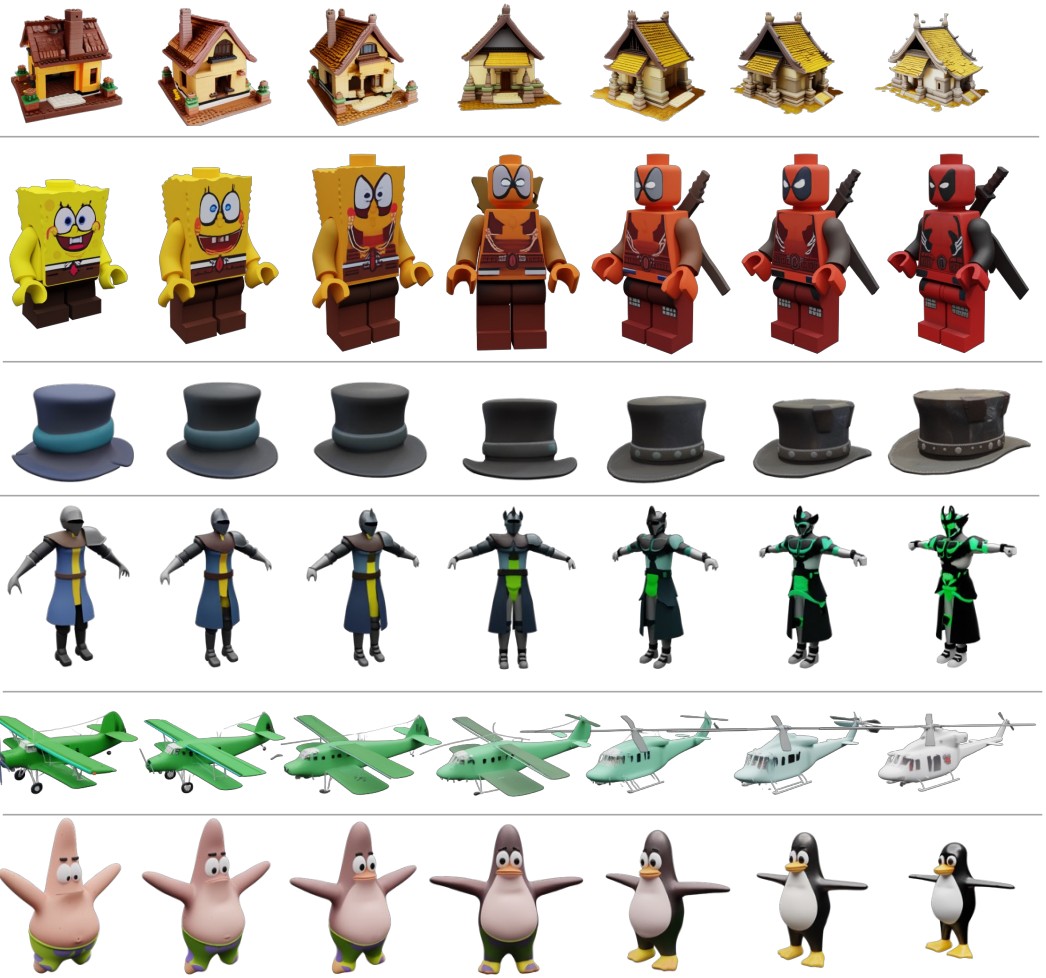

Figure 12: More qualitative results of Interp3D, which achieves fidelity and smooth transitions.

