# OpenReview forum: "Interp3D: Correspondence-aware Interpolation for Generative Textured 3D Morphing"
_ICLR.cc/2026/Conference — ICLR 2026 Poster_

### Official Review · Reviewer_vr3g · 2025-10-29

**Soundness:** 3
**Presentation:** 3
**Contribution:** 3
**Rating:** 6
**Confidence:** 4

**Summary:**

This paper proposes Interp3D, a novel and training-free framework for generative textured 3D morphing. The key insight is to address the critical challenge of 3D correspondence misalignments, a root cause of semantic ambiguity, structural distortion, and texture blurring in prior works. To this end, the authors introduce a progressive alignment framework that performs correspondence-aware interpolation across three levels: semantic (in condition space), structural (guided by SLAT features), and texture (via fine-grained fusion). Experiments on a newly constructed benchmark, Interp3DData, demonstrate that Interp3D outperforms existing baselines in terms of structural fidelity, transition smoothness, and visual plausibility.

**Strengths:**

1. Practical training-free formulation. The work presents a training-free method that effectively leverages pre-trained generative priors. This practical approach demonstrates strong performance, achieving superior results over existing baselines without the need for costly fine-tuning.
2. Contribution to the field via benchmarking. A key strength is the introduction of the Interp3DData benchmark. By categorizing the 3D morphing problem into distinct difficulty levels, the paper provides a valuable and standardized framework for evaluation, which facilitates future comparisons and advances in the field.
3. Clear motivation and well-structured methodology. The paper is clearly structured, with a well-articulated motivation that pinpoints the problem of 3D correspondence misalignments. The proposed three-stage progressive alignment framework is a logical and direct response to these identified challenges, making the technical contributions easy to follow.

**Weaknesses:**

1. Limited generality and model dependency. The entire proposed pipeline is intrinsically tied to the specific architecture and latent representations of the Trellis model. While this is a valid choice for a proof-of-concept, it raises questions about the generalizability of the method. The approach's strong dependency on Trellis-specific components (e.g., the SLAT representation) makes it non-trivial to adapt Interp3D to other 3D generative frameworks, thereby limiting its immediate applicability and scope.
2. Insufficient analysis of module interactions. The paper introduces three progressive alignment components but does not thoroughly investigate the interplay between them. For instance, it remains unclear whether the semantic alignment in the condition space and the subsequent SLAT-guided structure alignment always work in harmony or could potentially introduce conflicting signals in some cases. A more detailed ablation study, perhaps with counterexamples, is needed to demonstrate that this progressive strategy is consistently synergistic and does not lead to contradictory guidance during the generation process.

**Questions:**

1. In the semantic-aligned condition interpolation, the target condition tokens are permuted to match the source's semantic layout. While this benefits intermediate frames, could this permutation inadvertently harm the fidelity of the final generated target (when α=1)? At this extreme, one would expect an unaltered target output, but the permuted condition might distort it. Could the authors discuss or provide analysis on this potential issue and whether the alignment strategy is adjusted near the endpoints?
2. Given the potential for conflicting guidance between the different alignment modules (e.g., semantic vs. structural), could the authors provide further ablation studies that not only add modules progressively but also examine the performance of individual modules and other combinations? This would more directly validate the synergy of the proposed pipeline.

I would like to raise my score if above concerns are properly solved.

---

> ### Author Response · Authors · 2025-11-20
> **Response to Reviewer vr3g**
>
> > **Response to W1: Reliance on Trellis and Generalization.**
>
> Thanks for your feedback. Please refer to the **General Response for Common Concerns** part for our response.
>
> > **Response to W2 & Q2: About Verification for Module Interactions**
>
> Thanks for your suggestion.  We conducted detailed component analysis in Appendix A.4 (page.19)  to verify the module interactions. For convenience, we attach the ablation study below and analyze the relevant observations, in which A, B, C, D refer to initial condition interpolation as the base, semantic-aligned condition interpolation, SLAT-guided structure interpolation, and fine-grained texture fusion designs respectively.
>
>
> As shown in the Table below, applying SLAT-guided structure interpolation (C) alone contributes the most, reducing FID from 85.55 to 81.65 by explicitly stabilizing geometric transitions. In addition, combining semantic alignment (B) with texture fusion (D) yields 2.75 PPL and 0.097 LPIPS, indicating that high-level correspondence cues help maintain identity consistency, while fine-grained texture refinement suppresses local artifacts. Together, these results highlight that geometric coherence and appearance continuity are complementary factors driving overall perceptual quality.  The alignment modules operate synergistically.  Our coarse-to-fine design, semantic alignment establishes global correspondences, structural alignment refines geometry with 3D priors, and texture fusion transfers fine-grained appearance, operates on disjoint representation levels, their effects are complementary rather than conflicting.
>
> While the progressive design consistently improves morphing quality, we also examined rare extreme counterexamples. As shown in Figure 9 of Appendix A.0.4 (Page 16), adding texture fusion improves local color consistency and yields a more plausible transition for the  “boy → Santa Claus” case, but it also introduces slight appearance deviations from the endpoints, causing FID to increase from 103.69 to 107.0.
>
>
> | A    | B    | C    | D    | Easy FID↓ | Easy PPL↓ | Easy LPIPS↓ | Mid FID↓  | Mid PPL↓ | Mid LPIPS↓ | Hard FID↓ | Hard PPL↓ | Hard LPIPS↓ | Avg FID↓  | Avg PPL↓ | Avg LPIPS↓ |
> | ---- | ---- | ---- | ---- | --------- | --------- | ----------- | --------- | -------- | ---------- | --------- | --------- | ----------- | --------- | -------- | ---------- |
> | ✔    |      |      |      | 79.14     | 3.02      | 0.074       | 86.87     | 3.25     | 0.109      | 90.64     | 3.47      | 0.157       | 85.55     | 3.25     | 0.113      |
> | ✔    | ✔    |      |      | 75.09     | 2.75      | 0.067       | 86.82     | 2.98     | 0.101      | 88.63     | 3.24      | 0.146       | 83.51     | 2.99     | 0.105      |
> | ✔    |      | ✔    |      | 77.55     | 2.79      | 0.068       | 86.13     | 2.81     | 0.093      | 81.28     | 3.26      | 0.140       | 81.65     | 2.95     | 0.102      |
> | ✔    |      |      | ✔    | 77.37     | 2.68      | 0.065       | 84.89     | 3.03     | 0.101      | 86.71     | 3.16      | 0.144       | 82.99     | 2.95     | 0.104      |
> | ✔    | ✔    | ✔    |      | 73.81     | 2.67      | 0.066       | 84.64     | 2.61     | 0.088      | 86.42     | 3.21      | 0.143       | 81.62     | 2.83     | 0.098      |
> | ✔    | ✔    |      | ✔    | 75.58     | 2.49      | 0.061       | 86.89     | 2.81     | 0.095      | 85.55     | 2.94      | 0.134       | 82.67     | 2.75     | 0.097      |
> | ✔    | ✔    | ✔    | ✔    | **70.79** | **2.42**  | **0.059**   | **83.58** | **2.37** | **0.079**  | **82.54** | **2.62**  | **0.119**   | **78.97** | **2.47** | **0.086**  |
>
> >  **Response to Q1: About the Design for Matching Direction**
>
> Thanks for pointing this out. In our implementation,  the permutation is not applied at $\alpha_i=1$. The target is generated using its original condition features, identical to standard inference.The permutation is used only for $0 < \alpha_i < 1$, where we interpolate between the source and target generation processes, and apply alignments to improve correspondence in the intermediate frames.
>
> Regarding the concerns about distortions introduced by the permutations, our observations do not suggest deviation when near the endpoint.
> As presented in our visualized Figure 10 in Appendix A.0.5 (page 16), when $\alpha_i$ near 1, that the generated transition remains faithful, and the variations near the boundary do not affect the correctness of the final target output. An endpoint-aware variant (reversing the matching direction) still yields faithful results. This is expected, as the 3D generative prior stabilizes the endpoint, causing the model to converge to the canonical target regardless of alignment differences.

---

> > ### Comment · Reviewer_vr3g · 2025-11-24
> > **Thanks for rebuttal**
> >
> > I'd like to thank the authors for their rebuttal. I find the additional experiments on generalization and module interactions to be helpful. Although I believe that generalizing Interp3D to other 3D generation models remains nontrivial and requires considerable effort, which limits its applicability compared to other 3D morphing methods, the idea of leveraging generative model priors for 3D morphing is novel and interesting. Therefore, I have increased my score.

---

> ### Author Response · Authors · 2025-11-24
> **Reply to Reviewer vr3g**
>
> We sincerely appreciate your thoughtful assessment and the recognition for our work. Your positive recommendation is truly encouraging. We value your constructive suggestions and will revise our manuscript accordingly. Thank you again for your time and insightful feedback.

---

### Official Review · Reviewer_69AW · 2025-11-01

**Soundness:** 3
**Presentation:** 3
**Contribution:** 3
**Rating:** 6
**Confidence:** 5

**Summary:**

The paper proposes Interp3D, a training-free framework for textured 3D morphing that leverages the generative prior of TRELLIS. The method enforces correspondence progressively at three levels: (1) semantic-aligned condition interpolation in 2D token space, (2) SLAT-guided structure interpolation during 3D structure generation with fused attention and dynamic patch matching, and (3) fine-grained texture fusion for appearance transfer. The authors also curate Interp3DData and evaluate with FID, PPL, LPIPS plus a user study, showing consistent improvements over previous methods.

**Strengths:**

1. The method cleanly exploits TRELLIS to realize training-free 3D morphing, turning a strong generative prior into a controllable morph pipeline.

2. A dedicated benchmark (Interp3DData) and quantitative comparisons demonstrate that Interp3D surpasses prior SOTA on multiple metrics.

3. The paper is clearly structured. The presentation is clear and easy to follow.

4. Results are visually compelling—fewer geometric failures and significantly less texture blur than baselines.

5. User study included: Human preference indicates better fidelity, smoothness, and overall plausibility for Interp3D.

6. The paper computes alignment between condition tokens and generation-time tokens and then performs correspondence-aware attention interpolation, which is principled and well motivated.

**Weaknesses:**

Actually, I do not think this paper has an obvious weakness. My concern is about the manual hyperparameter sensitivity: The pipeline exposes numerous hand-tuned knobs (e.g., grid patch size and step-wise schedules in SLAT-Guided Structure Interpolation). It is unclear how robust these settings are across diverse source–target pairs or whether per-pair tuning is required.

**Questions:**

Failure cases: Do you observe failures when the source and target are semantically or structurally very far apart, or when TRELLIS reconstruction itself fails? Please include concrete examples, diagnostics, and whether the morph collapses or exhibits flickering/blur.

---

> ### Author Response · Authors · 2025-11-20
> **Response to Reviewer 69AW**
>
> We sincerely thank you for your appreciation and recognition for our work. Below, we provide detailed responses to the concerns.
>
> > **Response to W1: Hyperparameter Sensitivity**
>
> Thanks for your feedback. Please refer to the **General Response for Common Concerns** part for our response.
>
> >  **Response to Q1: Failure Case Analysis**
>
> Thanks for your valuable feedback. Yes, we do observe failures in extreme settings. As shown in Failure Case 1 of Figure 7 of Appendix A.0.2 (Page 15), the source and target are semantically and structurally too far apart. In such cases, no reliable correspondence can be established, and the morph tends to collapse toward one endpoint, resulting in abrupt transitions and sudden changes rather than a smooth trajectory.
>
> In Failure Case 2, we show an out-of-distribution example (e.g., sketched, flat 2D-like objects), where TRELLIS itself fails to reconstruct meaningful 3D geometry. Here, the morphing process degrades into collapsed geometry or heavy blur, since the 3D prior cannot provide a stable latent manifold to interpolate on.
>
> These failures are thus mainly attributable to the limitations of the underlying 3D generative prior under extreme semantic/structural gaps. Improving robustness in such challenging regimes, like applying stronger priors or adaptive fine-tuning can be important directions for future work.

---

### Official Review · Reviewer_qzgJ · 2025-11-01

**Soundness:** 3
**Presentation:** 3
**Contribution:** 3
**Rating:** 6
**Confidence:** 3

**Summary:**

Interp3D addresses this by coupling generative priors with reliable 3D correspondences through a progressive alignment principle.

SLAT-Guided Structure Interpolation: Enforces geometric consistency by maintaining structural correspondences using SLAT features from a 3D foundation model.

Fine-Grained Texture Fusion: Transfers appearance details by retrieving and fusing source and target features at corresponding locations, ensuring coherent and realistic surface appearance.

**Strengths:**

The paper presents a clear and detailed description of the Interp3D framework, including the three stages of alignment and the specific techniques used at each stage. The pseudocode provided in the appendix further enhances the clarity of the implementation.

The creation of the Interp3DData dataset provides a valuable resource for the research community.


The method is shown to be robust across a wide range of source–target pairs, including those with geometric and textural differences.

**Weaknesses:**

Current metrics (FID, PPL, LPIPS) focus on visual quality but lack assessment of semantic coherence and structural fidelity.

The method heavily relies on attention interpolation of an existing method Trellis3D, which lacks novelty. It's rather than revealing the findings of  Trellis3D.

**Questions:**

Will the Interp3DData data be open-sourced?

---

> ### Author Response · Authors · 2025-11-20
> **Response to Reviewer qzgJ**
>
> Thank you for your efforts and valuable feedback. Below, we provide detailed responses to the concerns.
>
> > **Response to W1: Evaluation Metrics on Semantic and Structure Fidelity.**
>
> Thank you for pointing this out. We introduce two additional aspects of metrics that explicitly evaluate the 3D structural fidelity and semantic coherence:
>
> 1. P-KID (Point-KID): We first extract PointNet features from 3D point samples of the generated intermediate shapes and the source/target shapes, and then compute a Kernel Inception Distance between these feature distributions. Lower P-KID indicates that the intermediate shapes stay closer to the structural manifold spanned by the endpoints, thus reflecting better 3D geometric fidelity.
>
> 2. CLIP-Dis/ CLIP-Sim: CLIP-based distance and similarity scores that quantify semantic alignment among the transition process. We use CLIP image features to measure the averaged adjacent-frame distance (CLIP-Dis, lower is better) and cosine similarity (CLIP-Sim, higher is better), capturing whether semantic evolution is smooth and coherent.
>
> As shown in the Table below, we evaluate both our method and prior morphing approaches under these new metrics on the whole Interp3DData, and Interp3D consistently achieves better semantic continuity and geometric stability, confirming that our improvements extend beyond visual quality.
>
>
> |                     | P-KID$\downarrow$ | CLIP-Dis$\downarrow$ | CLIP-Sim$\uparrow$ |
> | ------------------- | ----------------- | -------------------- | ------------------ |
> | DiffMorpher         | 0.6796            | 0.1253               | 0.8747             |
> | FreeMorph           | 0.5352            | 0.1034               | 0.8966             |
> | AID-I               | 0.4857            | 0.0651               | 0.9349             |
> | AID-O               | 0.5060            | 0.0603               | 0.9397             |
> | **Interp3D (Ours)** | **0.3961**        | **0.0529**           | **0.9471**         |
>
> **Table 1: Quantitative evaluation on semantic and structure fidelity metrics**
>
> -------
> > **Response to W2: About the Reliance and Design Novelty.**
>
>
> Thanks for your feedback.  For Interp3D, it is not relying on Trellis but a conceptual and targeted architectural solution to the core limitation of prior textured 3D morphing methods, which rely on 2D or latent interpolation and therefore ignore the correspondence misalignment. Instead of relying on Trellis attention,  Interp3D introduces the progressive 3D correspondence modeling: from semantic to 3D structural, and finally texture space, which directly addresses the alignment problem and preserves both geometry and appearance throughout the morph.
>
> We demonstrate this by applying the same correspondence mechanism to two additional 3D generative models, LN3Diff[1] and Topia3D-XL[2], leveraging their native structural embeddings as correspondence fields, and observing consistent gains in geometric stability and texture coherence across backbones, which is presented in Figure 6 of the manuscript Appendix A.0.1 (Page 14).
>
> -------
>
> >  **Response to Q1: About Data Open Source.**
>
> Yes, it will be open-sourced. We provide an anonymous download link in the Reproducibility Statement (page 10) of the manuscript, and it is accessible to all for verification.
>
> -------
>
> [1] Lan, Yushi, et al. "Ln3diff: Scalable latent neural fields diffusion for speedy 3d generation." European Conference on Computer Vision. 2024.
>
> [2] Chen, Zhaoxi, et al. "3dtopia-xl: Scaling high-quality 3d asset generation via primitive diffusion." Proceedings of the Computer Vision and Pattern Recognition Conference. 2025.

---

### Official Review · Reviewer_BzE3 · 2025-11-01

**Soundness:** 3
**Presentation:** 3
**Contribution:** 2
**Rating:** 4
**Confidence:** 4

**Summary:**

This paper tackles textured 3D morphing, aiming for smooth and plausible transitions between 3D assets that preserve both structure and appearance. The authors argue existing methods fail by either ignoring texture or by extending 2D strategies, which causes semantic and structural errors.

The paper proposes Interp3D, a training-free framework built on a "progressive alignment principle." This 3-stage process involves:

1. Semantic-Aligned Condition Interpolation: Matches semantic patches using DINOv2 features.

2. SLAT-Guided Structure Interpolation: Uses structured latents (SLAT) from a 3D model (TRELLIS) to guide geometric correspondence.

4. Fine-Grained Texture Fusion: Aggregates texture features to preserve detail.

A new dataset, Interp3DData, was built for evaluation. Results show Interp3D outperforms baselines in fidelity, smoothness, and plausibility.

**Strengths:**

1. The paper clearly identifies a specific technical problem (artifacts in 3D morphing) and proposes a logical, multi-stage framework that effectively solves it.

2. The training-free nature is a significant practical advantage, showing how to guide a generative prior using feature-space manipulation.

3. The evaluation is comprehensive, using quantitative metrics (FID, PPL, LPIPS) and a user study (Table 2) to prove its superiority over baselines.

4. The creation of Interp3DData is a useful, albeit minor, contribution to this specific sub-field.

**Weaknesses:**

1. The primary weakness is the limited scope and perceived significance of the task itself. 3D morphing is a relatively niche problem. The proposed solution, while effective, feels more like a clever engineering trick or application built on top of TRELLIS, rather than a novel, generalizable research contribution. The work is highly incremental.

2. The method seems tightly coupled to the TRELLIS model's SLAT representation. It's unclear if this principle generalizes to other 3D models.

3. The paper admits failure when semantic gaps are huge (Fig. 10), suggesting the initial DINOv2 patch matching is a bottleneck.

4. The "dynamic patch correspondence" (Sec 4.2) is vague. The paper lacks sensitivity analysis for the patch size $s_t$ schedule and the similarity threshold $\tau_0$.

**Questions:**

1. How generalizable is Interp3D? How dependent is it on the TRELLIS SLAT representation?

2. The failure case (Fig. 10) points to the DINOv2 matching as a bottleneck. Could this be improved with different 2D features (e.g., CLIP) or a 3D-native correspondence search?

3. Regarding Sec 4.2: What is the specific value for the threshold $\tau_0$ and the schedule used for decreasing the patch size $s_t$?

4. What is the performance gain from using the Beta(5, 5) distribution for sampling compared to standard linear interpolation?

---

> ### Author Response · Authors · 2025-11-20
> **Response to Reviewer BzE3**
>
> Thanks for your efforts and valuable suggestions. Below, we provide detailed responses to the concerns.
>
> > **Response to W1: Task Practicality and Design Novelty.**
>
> Thanks for your feedback. **For task practicality**, just as interpolation is essential in image and video generation, 3D morphing is crucial for creative applications involving 3D assets. In games and films, a common example is character evolution or shape-shifting effects, such as the transformations of heroes like the Hulk, Wukong, Mystique, or the T-1000. More importantly, these interpolated assets serve as the basis for 4D generation, like leveraging interpolation for upsampling [1], and the evolving of 3D intrinsics [2], which demonstrates its broad application for modern content creation.
>
> **For our design**, Interp3D is not an incremental extension but **a conceptual and targeted architectural solution to the core limitation of prior textured 3D morphing methods**, which rely on 2D or latent interpolation and therefore ignore the correspondence alignment. Interp3D instead introduces progressive 3D correspondence modeling: from semantic to 3D structural, and finally texture space, which directly addresses the alignment problem and preserves both geometry and appearance throughout the morph.
>
> Our design is **a general principle rather than a TRELLIS-specific trick**. We demonstrate this by applying Interp3D to two additional 3D generative models, LN3Diff[3] and 3DTopia-XL[4], leveraging their native structural embeddings as correspondence fields, and observing consistent gains in geometric stability and texture coherence across backbones, which is illustrated in the common response and the updated manuscript.
>
>
> > **Response to W2 and Q1: Reliance on Trellis and Its Generalization.**
>
> Thanks for your feedback. Please refer to the **General Response for Common Concerns** part for our response.
>
>
>
> >**Response to W3 and Q2: Failure Case Analysis and Attempts**
>
> Thank you for raising this point.  We politely clarify that this failure does not attributed to the failure of DINOv2. Rather, the extreme semantic gaps between the source and target leaves no meaningful correspondence to be established. This limitation persists regardless of the feature extractor used, as the underlying signals simply do not provide usable alignment cues.
>
> To verify this, we tested both reviewer-suggested alternatives, as shown in Figure 7 in Appendix A.0.2 (page 15). Attempt 1 and Attempt 2 replace our originally used features with CLIP-based 2D embeddings and 3D-native SLAT features, respectively, to estimate the correspondence. Although both alternatives produce slightly different local matches, neither resolves the underlying failure mode. The generated transitions remain structurally inconsistent, indicating that the limitation is not caused by the specific feature extractor but by the lack of reliable correspondences under large viewpoint or topology changes.
>
> For practical solutions, a promising way can be introducing artificial guidance. For example, a few user-defined anchors, an approximate UV hint, or a user-chosen intermediate shape to constrain the correspondence artificially.
>
>
> >  **Response to W4 and Q3: the Sensitive Analysis.**
>
> Thanks for your feedback. Please refer to the **General Response for Common Concerns** part for our response.
>
> > **Response to Q4: Gain from Using the Beta Distribution.**
>
> As analyzed in AID for 2D interpolation, linear interpolation with uniformly distributed interpolation coefficient $\alpha_i$ doesn’t yield a uniformly spaced distribution with smooth transitions, which is also revealed during our experiments in 3D generation. Thus, we apply the concave Beta(5, 5) distribution with a more shrinkage interpolation coefficient to the midpoint of the source and the target. As shown in the Figure 8 of Appendix A.0.3 (Page 15), Beta sampling yields more smoothly transformed intermediate shapes than uniform sampling. Quantitatively, it also achieves lower PPL and LPIPS, indicating smoother transitions and better perceptual coherence.
>
> | Method                | PPL ↓      | LPIPS ↓    |
> | --------------------- | ---------- | ---------- |
> | Uniform Distribution  | 2.2692     | 0.0582     |
> | **Beta Distribution** | **2.1072** | **0.0541** |
>
> -------
>
> [1] Ren, Jiawei, et al. "L4gm: Large 4d gaussian reconstruction model." Advances in Neural Information Processing Systems 37 (2024): 56828-56858.
>
> [2] Geng, Chen, et al. "Birth and Death of a Rose." Proceedings of the Computer Vision and Pattern Recognition Conference. 2025.
>
> [3] Lan, Yushi, et al. "Ln3diff: Scalable latent neural fields diffusion for speedy 3d generation." European Conference on Computer Vision. 2024.
>
> [4] Chen, Zhaoxi, et al. "3dtopia-xl: Scaling high-quality 3d asset generation via primitive diffusion." Proceedings of the Computer Vision and Pattern Recognition Conference. 2025.

---

### Author Response · Authors · 2025-11-20
**General Response for Common Concerns**

> **Interp3D's Generalization Ability.**

Interp3D is designed as a **model-agnostic concept rather than tied to Trellis or its SLAT representation**. The key design is that progressive 3D correspondence modeling guided by native feature cues from the source and target generation process, which can be applied for different 3D generation models.

To validate this, we implemented Interp3D across two additional 3D generation baselines, LN3Diff [1] and 3DTopia-XL [2]. Based on the new baselines, we first perform semantic-aware condition interpolation, and then instantiate the structural correspondence using each model’s native geometric embeddings (e.g., LN3Diff’s 3D latent tokens, 3DTopia’s primitive-level descriptors). These features provide stable geometric cues that let our structural alignment module plug in without modifying the backbone architectures. Finally, we apply our texture-refinement on their generative pipelines to complete the progressive alignment.

As shown in Figure 6 of Appendix A.0.1 (page 14), both approaches produce smooth and plausible morphings, confirming that Interp3D is a principled, architecture design rather than a model-specific enhancement.



> **Parameter Sensitive Analysis.**

We conduct ablations for the similarity threshold $\tau_0$ and the patch-size schedule $s_t$ on the hard cases of Inter3DData. We observe that the performance is robust across a reasonable range of thresholds. As shown in the Table below, $\tau_0=0.6$ achieves the best balance between stable matches and robustness with 82.13 FID and 2.6082 PPL.

For patch size $s_t$, we also conduct ablations on different starting patch size selections.  A linearly decreasing schedule beginning at $s_t=64$ produces the most consistent geometry. This matches the rectified-flow generation behavior: early steps favor large patches to stabilize coarse correspondences, while later steps benefit from smaller patches to refine fine-grained geometry.

The configuration corresponds to the **globally optimal setting** across the benchmark rather than per-pair tuning.


| $\tau_0$    | – (no use) | 0.5     | 0.6         | 0.7     | 0.8     |
| ----------- | ---------- | ------- | ----------- | ------- | ------- |
| **FID ↓**   | 83.5071    | 82.7629 | **82.1281** | 82.3760 | 82.3982 |
| **PPL ↓**   | 2.6107     | 2.6106  | **2.6082**  | 2.6118  | 2.6136  |
| **LPIPS ↓** | 0.1186     | 0.1183  | **0.1179**  | 0.1185  | 0.1186  |

**Table 1: Ablations on threshold selection $\tau_0$**

| $s_t$       | 1       | 8       | 27         | **64**      | 125     |
| ----------- | ------- | ------- | ---------- | ----------- | ------- |
| **FID ↓**   | 82.6624 | 82.3472 | 82.2640    | **82.1281** | 82.5908 |
| **PPL ↓**   | 2.6218  | 2.6100  | **2.5940** | 2.6082      | 2.6176  |
| **LPIPS ↓** | 0.1196  | 0.1187  | 0.1189     | **0.1182**  | 0.1184  |

**Table 2: Ablations on staring patch size $s_t$**

----------------


[1] Lan, Yushi, et al. "Ln3diff: Scalable latent neural fields diffusion for speedy 3d generation." *European Conference on Computer Vision*. 2024.

[2] Chen, Zhaoxi, et al. "3dtopia-xl: Scaling high-quality 3d asset generation via primitive diffusion." *Proceedings of the Computer Vision and Pattern Recognition Conference*. 2025.

---

### Author Response · Authors · 2025-11-20
**General Response for Manuscript Updated Information**

We sincerely thank all reviewers for their recognition and insightful feedback. We have updated the manuscripts with figures and related analysis in Appendix A.0 (starting on page 14). For convenience, titles of the newly added parts are highlighted in blue. The updated parts are organized in the following sections:

- Appendix A.0.1: Generalization Ability of Interp3D in Figure 6.
- Appendix A.0.2: Failure case analysis and Attempts in Figure 7.
- Appendix A.0.3: Gain from the Beta Distribution in Figure 8.
- Appendix A.0.4: Module Interaction Counterexamples in Figure 9.
- Appendix A.0.5: Correspondence Direction Analysis in Figure 10.

---

### Author Response · Authors · 2025-11-27

Dear Reviewers,

We sincerely thank all reviewers for your insightful comments and recognition of our work.

We also appreciate Reviewer vr3g for raising the score. As the discussion period deadline approaches, we would be grateful to know whether our rebuttal has adequately addressed your concerns. We are happy to provide any further clarification if needed.

Thank all reviewers again for the efforts and constructive feedback!

Best regards,

Author 1763

---

### Author Response · Authors · 2025-12-01
**Summary Response to ACs**

**Dear all ACs and Reviewers:**

We sincerely thank all reviewers for their efforts and constructive feedback throughout the discussion period. Here we provide a concise summary of our work, the reviewers’ recognitions, concerns, and our corresponding responses during the discussion phase.

------------

**Summary of Our Work:**

We propose **Interp3D**, a **training-free, correspondence-aware framework**  for textured 3D morphing, in with a **progressive alignment principle** is designed to jointly preserve geometric fidelity and texture coherence.

In our work:

- We analyze the challenges of existing textured 3D morphing paradigms and highlight the necessity to couple correspondence with the 3D generative priors for structural and textural consistency.
- We propose Interp3D, a training-free, correspondence-aware framework that integrates progressive semantic, structural, and texture alignment directly into the 3D generation pipeline, ensuring smooth and plausible transitions across the entire morphing trajectory.
- We curate Interp3DData with three difficulty levels and show that Interp3D significantly outperforms prior baselines, particularly on the hardest cases.

-------

**Reviewers’ Positive Feedback:**

During the discussion phase, we are encouraged by **three positive 6-rating** (Reviewer `qzgJ`, `69AW`, `vr3g`) and the reviewers’ recognition, which highlights:

- **Well-motivation and principled methodology,** that clearly tackles correspondence misalignment as the core limitation of prior textured 3D morphing methods and motivates the progressive alignment framework accordingly.
- **Practical, training-free design,** that effectively leverages generative priors without model fine-tuning.
- **A well-curated benchmark (Interp3DData),** that fills a missing evaluation gap in textured 3D morphing and enables standardized comparison across difficulty levels.
- **Clear presentation with solid quantitative results and human-study evidence**, supported by clear expression and figures, transparent implementation, detailed pseudocode, and comprehensive ablations.

------------------

**Responses to Reviewers’ Concerns:**

About the **Reviewers’ Concerns**, we have conducted additional experiments and analysis, including:

- **Generalization Beyond Trellis** (Reviewer `BzE3`, `vr3g`): We instantiated our Interp3D on LN3Diff and 3DTopia-XL baseline settings to demonstrate that our design is not Trellis-specific and generalizes well across 3D generative backbones.
- **Ablations on hyperparameter sensitivity and module interactions** (Reviewer `BzE3`, `69AW`, `vr3g`): We performed ablations on the threshold selection $\tau_0$, patch size setting $s_t$, and the interaction among the alignment modules, confirm robustness and complementary behavior among modules.
- **Failure Case Analysis** (Reviewer `BzE3`, `69AW`): We provided extreme failure cases and explain the causes, responding to the corresponding analysis and attempts for the solutions.
- **Additional Evaluation Metrics** (Reviewer `qzgJ`): We conducted experiments on additional evaluation metrics from semantic and 3D structure perspective for comprehensive qualitative results. This responds to requests for broader evaluations.
- **Detailed clarify and analysis:** We made detailed analysis on beta distribution (Reviewer `BzE3`), matching direction (Reviewer `vr3g`), and task practicality (Reviewer `BzE3`).

All corresponding numerical experiments, visualizations, and explanations are included in our discussion responses and the revised manuscript.

-----



Before 27 Nov 2025, Reviewer vr3g replied and found our response to be helpful. The overall ratings is raised from **(6, 6, 6, 4)** to **(8, 6, 6, 4)**.

We regret the impact of the OpenReview bug and hope this summary provides the Area Chair with a clear understanding of our work and the resolutions to the reviewers’ concerns. Thank you again for your time and efforts. If there are any remaining questions, we are happy to provide further clarification.

Best Regards,

The authors 1763.

---

### Meta-Review · Area_Chair_uif3 · 2026-01-06

**Summary:**

The paper presents a new method for training-free and correspondence-aware textured 3D morphing. It received mostly positive comments from the reviewers. The reviewers had concerns regarding the method’s generalization capability, the availability of training data, and the clarity of certain methodological details. The authors provided a comprehensive rebuttal to address these issues, with sufficient evidence from additional experiment results and analysis. One reviewer would like to increase the final score after considering the rebuttal. Overall, this paper presents an important contribution to this topic and achieves a significant performance level compared to previous SOTA results. Therefore, AC recommends acceptance of this submission.

**Reviewer Concerns:**

The reviewers have major concerns about the following aspects: the method’s generalization capability, the availability of training data, and the clarity of certain methodological details. The authors did an effective rebuttal to address these concerns.

**Reviewer Scores:**

The reviewer vr3g would like to raise the score from 6 to 8. So the final score distribution for this submission is 8, 6, 6, 4.

---

### Decision · Program_Chairs · 2026-01-26

Accept (Poster)